# Towards Universal Visual Reward and Representation via Value-Implicit Pre-Training

**Yecheng Jason Ma**[*2]**, Shagun Sodhani**[1]**, Dinesh Jayaraman**[2]**, Osbert Bastani**[2]**,
{Vikash Kumar**[†1]**, Amy Zhang**[†1]**}**

FAIR, Meta AI[1], University of Pennsylvania[2]

`https://sites.google.com/view/vip-rl`

## Abstract

Reward and representation learning are two long-standing challenges for learning an expanding set of robot manipulation skills from sensory observations. Given the inherent cost and scarcity of in-domain, task-specific robot data, learning from large, diverse, offline human videos has emerged as a promising path towards acquiring a generally useful visual representation for control; however, how these human videos can be used for general-purpose reward learning remains an open question. We introduce **V**alue-**I**mplicit **P**re-training (VIP), a self-supervised pre-trained visual representation capable of generating dense and smooth reward functions for unseen robotic tasks. VIP casts representation learning from human videos as an *offline goal-conditioned reinforcement learning* problem and derives a self-supervised goal-conditioned value-function objective that does not depend on actions, enabling pre-training on unlabeled human videos. Theoretically, VIP can be understood as a novel *implicit* time contrastive objective that generates a temporally smooth embedding, enabling the value function to be implicitly defined via the embedding distance, which can then be used to construct the reward function for any goal-image specified downstream task. Trained on large-scale Ego4D human videos and without any fine-tuning on in-domain, task-specific data, VIP can provide dense visual reward for an extensive set of simulated and **real-robot** tasks, enabling diverse reward-based visual control methods and outperforming all prior pre-trained representations. Notably, VIP can enable simple, *few-shot* offline RL on a suite of real-world robot tasks with as few as 20 trajectories.

## 1 Introduction

A long-standing challenge in robot learning is to develop robots that can learn a diverse and expanding set of manipulation skills from sensory observations (e.g., vision). This hope of developing general-purpose robots demands *scalable* and *generalizable* representation learning and reward learning to provide effective task representation and specification for downstream policy learning. Inspired by pre-training successes in computer vision (CV) (He et al., 2020; 2022) and natural language processing (NLP) (Devlin et al., 2018; Radford et al., 2019; 2021), pre-training visual representations on out-of-domain natural and human data (Deng et al., 2009; Grauman et al., 2022) has emerged as an effective solution for acquiring a general visual representation for robotic manipulation (Shah & Kumar, 2021; Parisi et al., 2022; Nair et al., 2022; Xiao et al., 2022) This paradigm is favorable to the traditional approach of in-domain representation learning because it does not require any intensive task-specific data collection or representation fine-tuning, and a single fixed representation can be used for a variety of unseen robotic domains and tasks (Parisi et al., 2022).

A key unsolved problem to pre-training for robotic control is the challenge of reward specification. Unlike simulated environments, real-world robotics tasks do not come with privileged environment state information or a well-shaped reward function defined over this state space. Prior pre-trained representations for control demonstrate results in only visual reinforcement learning (RL) in simulation, assuming access to a well-shaped dense reward function (Shah & Kumar, 2021; Xiao et al., 2022), or

---

[*]Corresponding author: `jasonyma@seas.upenn.edu`. [†] equal advising, randomized order.

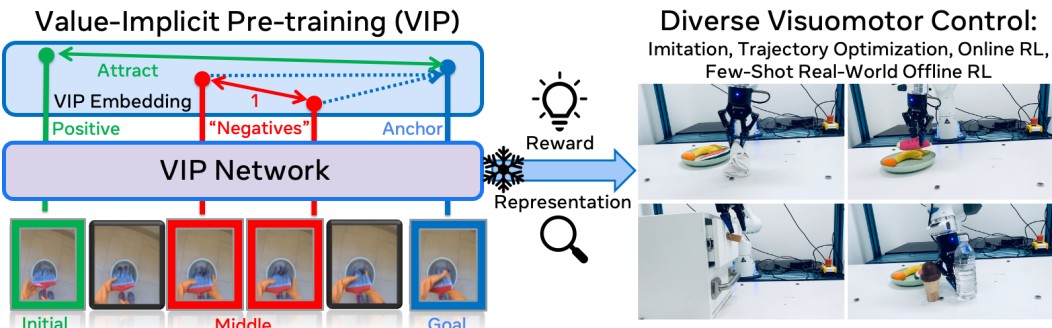

Figure 1: **Value-Implicit Pre-training (VIP)**. Pre-trained on large-scale, in-the-wild human videos, frozen VIP network can provide visual reward and representation for downstream unseen robotics tasks and enable diverse visuomotor control strategies without any task-specific fine-tuning.

visual imitation learning (IL) from demonstrations (Parisi et al., 2022; Nair et al., 2022). In either case, substantial engineering effort is required for learning each new task. Instead, a simple and general way of specifying real-world manipulation tasks is by providing a goal image (Andrychowicz et al., 2017; Pathak et al., 2018) that captures the desired visual changes to the environment. However, as we demonstrate in our experiments, existing pre-trained visual representations do not produce effective reward functions in the form of embedding distance to the goal image, despite their effectiveness as pure visual encoders. Given that these models are already some of the most powerful models derived from computer vision, it begs the pertinent question of whether a *universal* visual reward function learned entirely from out-of-domain data is even possible.

In this paper, we show that such a general reward model can indeed be derived from a pre-trained visual representation, and we acquire this representation by treating representation learning from diverse human-video data as a big *offline goal-conditioned reinforcement learning* problem. Our idea philosophically diverges from all prior works: instead of taking what worked the best for CV tasks and "hope for the best" in visual control, we propose a more principled approach of using reinforcement learning itself as a pre-training mechanism for reinforcement learning. Now, this formulation certainly seems impractical at first because human videos do not contain any action information for policy learning. Our key insight is that instead of solving the impossible *primal* problem of direct policy learning from out-of-domain, action-free videos, we can instead solve the *Fenchel dual* problem of goal-conditioned value function learning. This dual value function, as we will show, can be trained without actions in an entirely self-supervised manner, making it suitable for pre-training on (out-of-domain) videos without robot action labels.

Theoretically, we show that this dual objective amounts to a novel form of *implicit* time contrastive learning, which attracts the representations of the initial and goal frame in the same trajectory, while implicitly repelling the representations of intermediate frames via recursive one-step temporal-difference minimization. These properties enable the representation to capture long-range temporal dependencies over distant task frames and inject local temporal smoothness over neighboring frames, making for smooth embedding distances that we show are the key ingredient of an effective reward function. This contrastive lens, importantly, enables the value function to be implicitly defined as a similarity metric in the embedding space, resulting in our simple final algorithm, **V**alue-**I**mplicit **P**re-training (VIP); see Fig. 1 for an overview.

Trained on the large-scale, in-the-wild Ego4D human video dataset (Grauman et al., 2022) using a simple sparse reward, VIP is able to capture a general notion of goal-directed task progress that makes for effective reward-specification for unseen robot tasks specified via goal images. On an extensive set of simulated and real-robot tasks, VIP's visual reward significantly outperforms those of prior pre-trained representations on a diverse set of reward-based policy learning paradigms. Coupled with a standard trajectory optimizer (Williams et al., 2017), VIP can solve $\approx 30\%$ of the tasks without any task-specific hyperparameter or representation fine-tuning and is the only representation that enables non-trivial progress on the set of more difficult tasks. Given more optimization budget, VIP's performance can further improve up to $\approx 45\%$, whereas other representations do worse due to reward hacking. When serving as *both* the visual representation and reward function for visual online RL, VIP again significantly outperforms prior methods by a wide-margin and achieves 40% aggregate success rate. To the best of our knowledge, these are the first demonstrations of a successful

perceptual reward function learned using entirely out-of-domain data. Finally, we demonstrate that VIP can enable real-world *few-shot* offline RL with as few as 20 trajectories on a diverse suite of real-robot manipulation tasks, demonstrating for the first time that offline RL is possible in this low-data regime and paving the way towards truly scalable and autonomous robot learning.

## 2 RELATED WORK

We review relevant literature on (1) Out-of-Domain Representation Pre-Training for Control, (2) Perceptual Reward Learning from Human Videos, and (3) Goal-Conditioned RL as Representation Learning. Due to space constraint, the latter two are included in App. B.

**Out-of-Domain Representation Pre-Training for Control.** Bootstrapping visual control using frozen representations pre-trained on out-of-domain non-robot data is a nascent field that has seen fast progress over the past year. Shah & Kumar (2021) demonstrates that pre-trained ResNet (He et al., 2016) representation on ImageNet (Deng et al., 2009) serves as effective visual backbone for simulated dexterous manipulation RL tasks. Parisi et al. (2022) finds ResNet models trained with unsupervised objectives, such as momentum contrastive learning (MOCO) (He et al., 2020), to surpass supervised objectives (e.g, image classification) for both visual navigation and control tasks. Xiao et al. (2022) demonstrates that masked-autoencoder (He et al., 2022) trained on diverse video data (Goyal et al., 2017; Shan et al., 2020) can be an effective visual embedding for online RL. The closest work to ours is R3M Nair et al. (2022), which is also pre-trained on the Ego4D dataset and attempts to capture temporal information in the videos by using time-contrastive learning (Sermanet et al., 2018); whereas VIP is fully self-supervised, R3M additionally requires video textual descriptions to align its representation. These prior works primarily re-purpose existing objectives and models for visual control and do not address the reward specification challenge. In contrast, VIP is the first to propose a novel RL-based objective for out-of-domain pre-training and is capable of producing generalizable dense reward signals that enable several new visuomotor control strategies that have not been demonstrated in this setting before.

## 3 PROBLEM SETTING AND BACKGROUND

In this section, we describe our problem setting of out-of-domain pre-training and provide formalism for downstream representation evaluation. Additional background on goal-conditioned reinforcement learning and contrastive learning is included in App. A.

**Out-of-Domain Pre-Training Visual Representation.** We assume access to a training set of video data $D = \{v_i := (o_1^i, ..., o_{h_i}^i)\}_{i=1}^N$, where each $o \in O := \mathbb{R}^{H \times W \times 3}$ is a raw RGB image; note that this formalism also captures standard image datasets (e.g., ImageNet), if we take $h_i = 1$ for all $v_i$. Like prior works, we assume $D$ to be out-of-domain and does not include any robot task or domain-specific data. A learning algorithm $\mathcal{A}$ ingests this training data and outputs a visual encoder $\phi := \mathcal{A}(D) : \mathbb{R}^{H \times W \times 3} \to K$, where $K$ is the embedding dimension.

**Representation Evaluation.** Given a choice of representation $\phi$, every evaluation task can be instantiated as a Markov decision process $\mathcal{M}(\phi) := (\phi(O), A, R(o_t, o_{t+1}; \phi, g), T, \gamma, g)$, in which the state space is the induced space of observation embeddings, and the task is specified via a (set of) goal image(s) $g$. Specifically, for a given transition tuple $(o_t, o_{t+1})$, we define the reward to be the goal-embedding distance difference (Lee et al., 2021; Li et al., 2022):

$$R(o_t, o_{t+1}; \phi, \{g\}) := \mathcal{S}_\phi(o_{t+1}; g) - \mathcal{S}_\phi(o_t; g) := (1 - \gamma)\mathcal{S}_\phi(o_{t+1}; g) + (\gamma \mathcal{S}_\phi(o_{t+1}; g) - \mathcal{S}_\phi(o_t; g)),$$
(1)

where $\mathcal{S}_\phi$ is a distance function in the $\phi$-representation space; in this work, we set $\mathcal{S}_\phi(o_t; g) := -\|\phi(o_t) - \phi(g)\|_2$. This reward function can be interpreted as a raw embedding distance reward with a reward shaping (Ng et al., 1999) term that encourages making progress towards the goal. This preserves the optimal policy but enables more efficient and robust policy learning.

Under this formalism, parameters of $\phi$ are frozen during policy learning (it is considered a part of the MDP), and we want to learn a policy $\pi : \mathbb{R}^K \to A$ that outputs an action based on the embedded observation $a \sim \pi(\phi(o))$. Note that while our evaluation tasks are defined via visual goals, the policies do not explicit condition on a goal as in multi-goal RL (Andrychowicz et al., 2017) setting because there is only one goal to be attempted per task.

## 4 VALUE-IMPLICIT PRE-TRAINING

In this section, we demonstrate how a self-supervised value-function objective can be derived from computing the dual of an offline RL objective on passive human videos (Section 4.1). Then, we show how this objective amounts to a novel implicit formulation of temporal contrastive learning (Section 4.2), which helps inducing a temporally smooth embedding favorable for downstream visual reward specification. Finally, we leverage this contrastive interpretation to instantiate a simple implementation (<10 lines of PyTorch code) of our dual value objective that does not explicitly learn a value network (Section 4.3), culminating in our final algorithm, Value-Implicit Pre-training (VIP).

### 4.1 FOUNDATION: SELF-SUPERVISED VALUE LEARNING FROM HUMAN VIDEOS

While human videos are out-of-domain data for robots, they are *in-domain* for learning a goal-conditioned policy $\pi_H$ over human actions, $a^H \sim \pi^H(\phi(o) \mid \phi(g))$, for some human action space $A^H$. Therefore, given that human videos naturally contain goal-directed behavior, one reasonable idea of utilizing offline human videos for representation learning is to solve an offline goal-conditioned RL problem over the space of human policies and then extract the learned visual representation. To this end, we consider the following KL-regularized offline RL objective (Nachum et al., 2019) for some to-be-specified reward $r(o, g)$:

$$\max_{\pi_H, \phi} \mathbb{E}_{\pi^H} \left[ \sum_t \gamma^t r(o; g) \right] - D_{\mathrm{KL}}(d^{\pi_H}(o, a^H; g) \| d^D(o, \tilde{a}^H; g)), \tag{2}$$

where $d^{\pi_H}(o, a^H; g)$ is the distribution over observations and actions $\pi_H$ visits conditioned on $g$. Observe that a "dummy" action $\tilde{a}$ is added to every transition $(o_h^i, \tilde{a}_h^i, o_{h+1}^i)$ in the dataset $D$ so that the KL regularization is well-defined, and $\tilde{a}_i^h$ can be thought of as the unobserved *true* human action taken to transition from observation $o_h^i$ to $o_{h+1}^i$. While this objective is mathematically sound and encourages learning a conservative $\pi^H$, it is seemingly implausible because the offline dataset $D^H$ does not come with any action labels nor can $A^H$ be concretely defined in practice. However, what this objective does provide is an elegant *dual* objective over a value function that does not depend on any action label in the offline dataset. In particular, leveraging the idea of Fenchel duality (Rockafellar, 1970) from convex optimization, we have the following result:

**Proposition 4.1.** *Under assumption of deterministic transition dynamics, the dual optimization problem of equation 2 is*

$$\max_{\phi} \min_V \mathbb{E}_{p(g)} \left[ (1 - \gamma) \mathbb{E}_{\mu_0(o; g)}[V(\phi(o); \phi(g))] + \log \mathbb{E}_{(o, o'; g) \sim D} \left[ \exp\left( r(o, g) + \gamma V(\phi(o'); \phi(g)) - V(\phi(o), \phi(g)) \right) \right] \right], \tag{3}$$

*where $\mu_0(o; g)$ is the goal-conditioned initial observation distribution, and $D(o, o'; g)$ is the goal-conditioned distribution of two consecutive observations in dataset $D$.*

As shown, actions do not appear in the objective. Furthermore, since all expectations in equation 3 can be sampled using the offline dataset, this dual value-function objective can be self-supervised with an appropriate choice of reward function. In particular, since our goal is to acquire a value function that extracts a general notion of goal-directed task progress from passive offline human videos, we set $r(o, g) = \mathbb{I}(o == g) - 1$, which we refer to as $\tilde{\delta}_g(o)$ in shorthand. This reward provides a constant negative reward when $o$ is not the provided goal $g$, and does not require any task-specific engineering. The resulting value function $V(\phi(o); \phi(g))$ captures the discounted total number of steps required to reach goal $g$ from observation $o$. Consequently, the overall objective will encourage learning visual features $\phi$ that are amenable to predicting the discounted temporal distance between two frames in a human video sequence. With enough size and diversity in the training dataset, we hypothesize that this value function can generalize to completely unseen (robot) domains and tasks.

### 4.2 ANALYSIS: IMPLICIT TIME CONTRASTIVE LEARNING

While equation 3 will learn some useful visual representation via temporal value function optimization, in this section, we show that it can be understood as a novel *implicit* temporal contrastive learning objective that acquires temporally smooth embedding distance over video sequences, underpinning VIP's efficacy jointly as a visual representation and reward for downstream control.

We begin by simplifying the expression in equation 3 by first assuming that the optimal $V^*$ is found:

$$\min_\phi \mathbb{E}_{p(g)}\left[(1-\gamma)\mathbb{E}_{\mu_0(o;g)}[-V^*(\phi(o);\phi(g))] + \log\mathbb{E}_{D(o,o';g)}\left[\exp\left(\tilde{\delta}_g(o) + \gamma V(\phi(o');\phi(g)) - V(\phi(o),\phi(g))\right)\right]^{-1}\right], (4)$$

where we have also re-written the maximization problem as a minimization problem. Now, after few algebraic manipulation steps (see App. C for a derivation), if we think of $V^*(\phi(o);\phi(g))$ as a *similarity metric* in the embedding space, then we can massage equation 4 into an expression that resembles the InfoNCE (Oord et al., 2018) time contrastive learning (Sermanet et al., 2018) (see App. A.2 for a definition and additional background) objective:

$$\min_\phi (1-\gamma)\mathbb{E}_{p(g),\mu_0(o;g)}\left[-\log\frac{e^{V^*(\phi(o);\phi(g))}}{\mathbb{E}_{D(o,o';g)}\left[\exp\left(\tilde{\delta}_g(o) + \gamma V^*(\phi(o');\phi(g)) - V^*(\phi(o),\phi(g))\right)\right]^{\frac{-1}{(1-\gamma)}}}\right] (5)$$

In particular, $p(g)$ can be thought of the distribution of "anchor" observations, $\mu_0(s;g)$ the distribution of "positive" samples, and $D(o,o';g)$ the distribution of "negative" samples. Counter-intuitively and in contrast to standard single-view time contrastive learning (TCN), in which the positive observations are temporally closer to the anchor observation than the negatives, equation 5 has the positives to be as temporally far away as possible, namely the initial frame in the the same video sequence, and the negatives to be middle frames sampled in between. This departure is accompanied by the equally intriguing deviation of the lack of explicit repulsion of the negatives from the anchor; instead, they are simply encouraged to minimize the (exponentiated) one-step temporal-difference error in the representation space (the denominator in equation 5); see Fig. 1. Now, since the value function encodes negative discounted temporal distance, due to the recursive nature of value temporal-difference (TD), in order for the one-step TD error to be globally minimized along a video sequence, observations that are temporally farther away from the goal will naturally be repelled farther away in the representation space compared to observations that are nearby in time; in App. C.3, we formalize this intuition and show that this repulsion always holds for optimal paths. Therefore, the repulsion of the negative observations is an *implicit*, emergent property from the optimization of equation 5, instead of an explicit constraint as in standard (time) contrastive learning.

Now, we dive into why this *implicit* time contrastive learning is desirable. First, the explicit attraction of the initial and goal frames enables capturing *long-range* semantic temporal dependency as two frames that meaningfully indicate the beginning and end of a task are made close in the embedding space. This closeness is also well-defined due to the one-step TD backup that makes every embedding distance recursively defined to be the discounted number of timesteps to the goal frame. Combined with the implicit yet structured repulsion of intermediate frames, this push-and-pull mechanism helps inducing a *temporally smooth* and consistent representation.

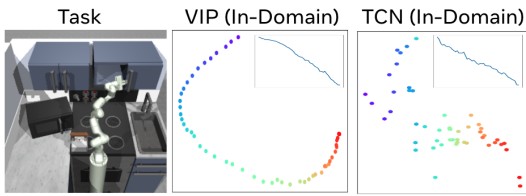

Figure 2: Learned 2D representation of a held-out task demonstration by VIP and TCN trained on task-specific in-domain data. The color gradient indicates trajectory time progression (purple for beginning, red for end). The inset plots are embedding distances to the last frame.

In particular, as we pass a video sequence in the training set through the trained representation, the embedding should be structured such that two trends emerge: (1) neighboring frames are close-by in the embedding space, (2) their distances to the last (goal) frame smoothly decrease due to the recursively defined embedding distances. To validate this intuition, in Fig. 2, we provide a simple toy example comparing implicit vs. standard time contrastive learning when trained on *in-domain, task-specific* demonstrations; details are included in App. E.2. As shown, standard time contrastive learning only enforces a coarse notion of temporal consistency and learns a non-locally smooth representation that exhibits many local minima. In contrast, VIP learns a much better structured embedding that is indeed temporally consistent and locally smooth.

### 4.3 Algorithm: Value-Implicit Pre-Training (VIP)

Recall that $V^*$ is assumed to be known for the derivation in Section 4.2, but in practice, its analytical form is rarely known. Now, given that $V^*$ plays the role of a distance measure in our implicit time contrastive learning framework, a simple and practical way to approximate $V^*$ is to set it to be a

choice of similarity metric, bypassing having to explicitly parameterize it as a neural network. In this work, we choose the common choice of the negative $L_2$ distance used in prior work Sermanet et al. (2018); Nair et al. (2022): $V^*(\phi(o), \phi(g)) := - \|\phi(o) - \phi(g)\|_2$. Given this choice, our final representation learning objective is as follows:

$$L(\phi) = \mathbb{E}_{p(g)} \left[ (1-\gamma)\mathbb{E}_{\mu_0(o;g)} \left[ \|\phi(o) - \phi(g)\|_2 \right] + \log \mathbb{E}_{(o,o';g) \sim D} \left[ \exp\left( \|\phi(o) - \phi(g)\|_2 - \tilde{\delta}_g(o) - \gamma \|\phi(o') - \phi(g)\|_2 \right) \right] \right], \quad (6)$$

in which we also absorb the exponent of the log-sum-exp term in 4 into the inner $\exp(\cdot)$ term via Jensen's inequality; we found this upper bound to be numerically more stable. To sample video trajectories from $D$, because any sub-trajectory of a video is also a valid video sequence, VIP samples these sub-trajectories and treats their initial and last frames as samples from the goal and initial-state distributions (Step 3 in Alg. 1). Altogether, VIP training is illustrated in Alg. 1; it is simple and its core training loop can be implemented in fewer than 10 lines of PyTorch code (Alg. 2 in App. D.3).

---

**Algorithm 1** Value-Implicit Pre-Training (VIP)

1: **Require**: Offline (human) videos $D = \{(o_1^i, ..., o_{h_i}^i)\}_{i=1}^N$, visual architecture $\phi$
2: **for** number of training iterations **do**
3:     Sample sub-trajectories $\{o_t^i, ..., o_k^i, o_{k+1}^i, ..., o_T^i\}_{i=1}^B \sim D, t \in [1, h_i - 1], t \leq k < T, T \in (t, h_i], \forall i$
4:     $\mathcal{L}(\phi) := \frac{1-\gamma}{B} \sum_{i=1}^B \left[ \|\phi(o_t^i) - \phi(o_T^i)\|_2 \right] + \log \frac{1}{B} \sum_{i=1}^B \left[ \exp\left( \|\phi(o_k^i) - \phi(o_T^i)\|_2 - \tilde{\delta}_{o_T^i}(o_k^i) - \gamma \|\phi(o_{k+1}^i) - \phi(o_T^i)\|_2 \right) \right]$
5:     Update $\phi$ using SGD: $\phi \leftarrow \phi - \alpha_\phi \nabla\mathcal{L}(\phi)$

---

## 5 EXPERIMENTS

In this section, we demonstrate VIP's effectiveness as both a pre-trained visual reward and representation on three distinct reward-based policy learning settings. We begin by detailing VIP's training and introducing baselines. Then, we present the full evaluation results for each setting, and we conclude with qualititave analysis, delving into VIP's unique effectiveness.

**VIP Training.** We use a standard ResNet50 (He et al., 2016) architecture as VIP's visual backbone and train on a subset of the Ego4D dataset (Grauman et al., 2022), a large-scale egocentric video dataset consisting of humans accomplishing diverse tasks around the world. These choices are identical to a prior work (Nair et al., 2022); additionally, we use the exact same hyperparameters (e.g., batch size, optimizer, learning rate) as in Nair et al. (2022). See App. D for details.

**Baselines.** The closest comparison is **R3M** (Nair et al., 2022), which pre-trains on the same Ego4D dataset using a combination of time contrastive learning, $L_1$ weight regularization, and language embedding consistency losses. We also consider a self-supervised ResNet50 network trained on ImageNet using Momentum Contrastive (**MoCo**), a supervised **ResNet**50 network trained on ImageNet, and **CLIP** Radford et al. (2021), covering a wide range of pre-existing visual representations that have been

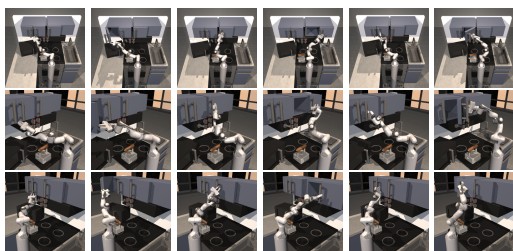

Figure 3: Frankakitchen example goal images.

used for robotics control (Shah & Kumar, 2021; Parisi et al., 2022; Cui et al., 2022), though none has been tested in our three reward-based settings in which the reward also has to be produced by the representation. Note that the visual backbone for all methods is ResNet50, enabling a fair comparison of the pre-training objectives. Besides VIP, all models are taken from their publicly released checkpoints. In Appendix G.1, we additionally compare to MoCo and Masked Auto-Encoder (MAE) models trained on Ego4D.

**Evaluation Environments** For our simulation experiments, we consider the FrankaKitchen (Gupta et al., 2019) environment, in which a 7-DoF Franka robot is tasked with manipulating common household kitchen objects to pre-specified configurations. We use all 12 subtasks supported in the environment and 3 camera views (left, center, right) for each task, in total of 36 visual manipulation tasks. Furthermore, we consider two initial robot state for every task, one Easy setting in which the end-effector is initialized close to the object of interest, and one Hard setting in which the end-effector is uniformly initialized above the stove edge regardless of the task. The task horizon is 50 (resp. 100)

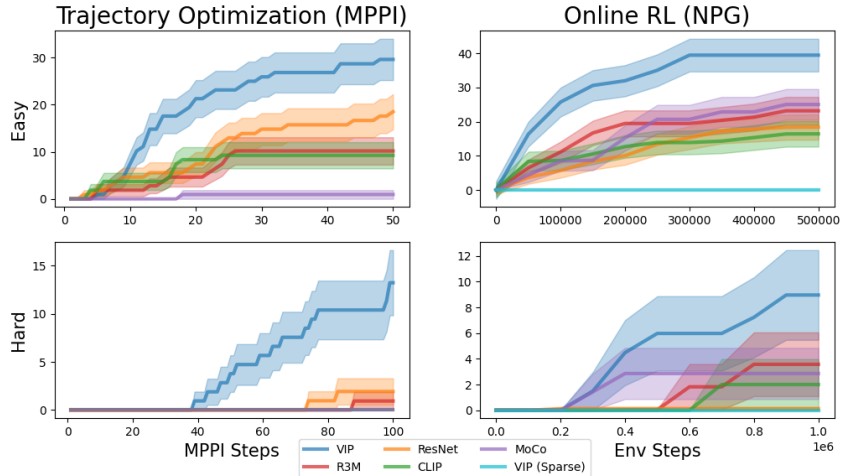

Figure 4: Visual trajectory optimization and online RL aggregate results (cumulative success rate %).

for the Easy (resp. Hard) setting. Each task is specified via a goal image; see Fig. 3 for example goal images for all three views, and see Fig. 8-9 in App. E.1 for initial and goal frames for all tasks).

## 5.1 TRAJECTORY OPTIMIZATION & ONLINE REINFORCEMENT LEARNING

We evaluate pre-trained representations' capability as pure visual reward functions by using them to directly synthesize a sequence of actions using trajectory optimization In particular, we use **model-predictive path-integral (MPPI)** (Williams et al., 2017). To evaluate each proposed sequence of actions, we directly roll it out in the simulator for simplicity. Vanilla trajectory optimization, though sample efficient, is prone to local minima in the reward landscape due to a lack of trial-and-error exploration. We hypothesize that online RL may be able to overcome bad local minima, but it comes with the added challenge of demanding the pre-trained representation to provide both the visual reward and representation for learning a closed-loop policy. To further study the importance of a learned dense reward in online RL, we compare to using ground-truth task sparse reward coupled with VIP's visual representation, **VIP (Sparse)**. The RL algorithm we use is **natural policy gradient (NPG)** (Kakade, 2001). We leave all experiment details are in App. E.3-E.4. In Fig. 4, we report each representation's cumulative success rate averaged over all configurations (3 seeds * 3 cameras * 12 tasks = 108 runs); the success rate for online RL is computed over a separate set of test rollouts.

Examining the MPPI results, we see that VIP is substantially better than all baselines in both Easy and Hard settings, and is the only representation that makes non-trivial progress on the Hard setting. In Fig. 5, we couple VIP and the strongest baselines (R3M, Resnet) with increasingly more powerful MPPI optimizers (i.e., more trajectories per optimization step; default 32 are used in Fig. 4). As shown, while VIP steadily benefits from stronger optimizers and can reach an average success rate of **44%**, baselines often do *worse* when MPPI is given more compute budget, suggesting that their reward landscapes are filled with local minima that do not correlate with task progress and are easily exploited by stronger optimizers. To further validate these observations, in App. E.3, we report the $L_2$ error to

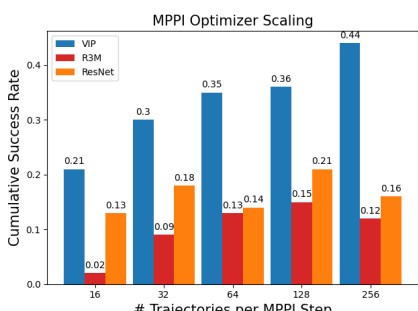

Figure 5: VIP benefits from compute scaling in downstream control.

the ground-truth goal-image robot and object poses after each environment step taken by MPPI. We find that VIP is able to minimize both the robot and object pose errors quite robustly over diverse tasks and views, whereas several baselines in fact *increase* the robot pose error on average. Given these findings, we hypothesize that VIP's reward functions are able capture task-salient information in the visual observations. In App. G.4, we validate this hypothesis and find that on at least one camera view for 8 out of the 12 tasks, VIP's rewards are highly correlated with the human-engineered state-based dense rewards, with correlation coefficients as high as $R^2 = \mathbf{0.95}$, highlighting its potential of replacing manual reward engineering without any prior knowledge about the robot domain or tasks.

Switching gears to online RL, VIP again achieves consistently superior performance. VIP (Sparse)'s inability to solve any task, despite a strong visual representation provided by VIP itself, indicates

Table 1: Real-robot offline RL results (success rate % averaged over 10 rollouts with standard deviation reported).

| Environment | Pre-Trained VIP-RWR | VIP-BC | R3M-RWR | R3M-BC | In-Domain Scratch-BC | VIP-RWR | VIP-BC |
|---|---|---|---|---|---|---|---|
| CloseDrawer | $\mathbf{100} \pm 0$ | $50 \pm 50$ | $80 \pm 40$ | $10 \pm 30$ | $30 \pm 46$ | $0 \pm 0$ | $0^* \pm 0$ |
| PushBottle | $\mathbf{90} \pm 30$ | $50 \pm 50$ | $70 \pm 46$ | $50 \pm 50$ | $40 \pm 48$ | $0^* \pm 0$ | $0^* \pm 0$ |
| PlaceMelon | $\mathbf{60} \pm 48$ | $10 \pm 30$ | $0 \pm 0$ | $0 \pm 0$ | $0 \pm 0$ | $0^* \pm 0$ | $0^* \pm 0$ |
| FoldTowel | $\mathbf{90} \pm 30$ | $20 \pm 40$ | $0 \pm 0$ | $0 \pm 0$ | $0 \pm 0$ | $0^* \pm 0$ | $0^* \pm 0$ |

the necessity of dense reward in solving these challenging visual manipulation tasks and further accentuates VIP's versatility doubling as both visual reward and representation; furthermore, we find that sparse reward coupled with even the true state representation is unable to make any progress. Finally, we comment that whereas sparse reward still requires human engineering via installing additional sensors (Rajeswar et al., 2021; Singh et al., 2019) and faces exploration challenges (Nair et al., 2018), with VIP, the end-user has to provide only a goal image.

## 5.2 REAL-WORLD FEW-SHOT OFFLINE REINFORCEMENT LEARNING

In this section, we demonstrate how VIP's reward and representation can power a simple and practical system for real-world robot learning in the form of *few-shot* offline reinforcement learning, making offline RL simple, sample-efficient, and more effective than BC with almost no added complexity.

To this end, we consider a simple reward-weighted regression (RWR) (Peters & Schaal, 2007; Peng et al., 2019) approach, in which the reward and the encoder are provided by the pre-trained model $\phi$:

$$\mathcal{L}(\pi) = -\mathbb{E}_{D_{\text{task}}(o,a,o',g)} \left[ \exp(\tau \cdot R(o, o'; \phi, g)) \log \pi(a \mid \phi(o)) \right], \tag{7}$$

where $R$ is defined via equation 1 and $\tau$ is the temperature scale. Compared to BC, which would be equation 7 with uniform weights to all transitions, RWR can focus policy learning on transitions that have high rewards (i.e., high task progress) under the deployed representation. Consequently, if the chosen representation is predictive of task progress (i.e., assigning high rewards to key transitions for the task), then RWR should be able to outperform BC, especially on tasks that naturally admit key transitions (e.g., picking up the towel edge in our FoldTowel task). This intuition is theoretically proven (Kumar et al., 2022) and holds even if the offline data consists of solely expert demonstrations, though not validated on real-world tasks.

We introduce 4 tabletop manipulation tasks (see Fig. 1 and Fig. 13) requiring a real 7-DOF Franka robot to manipulate objects drawn from distinct categories of objects. For each task, we collect in-domain, task-specific offline data $D_{\text{task}}$ of $\sim 20$ demonstrations with randomized object initial placements for policy learning; we provide detailed task and experiment descriptions in App. F. We compare VIP to R3M, the only other representation that has been deployed on real robots in the visual imitation setting, and instantiate **{VIP,R3M}-{RWR,BC}**. To assess whether pre-training is necessary for low-data regime offline RL, we also train **in-domain** VIP-{RWR,BC} from scratch using only $D_{\text{task}}$, where the VIP representation is learned using $D_{\text{task}}$ first and then frozen during the policy training via RWR/BC. In addition, we include **Scratch-BC**, for which the regression loss is used to learn the policy and the representation in a completely end-to-end manner.

We train all policies using the same set of hyperparameters used for real-world BC training in Nair et al. (2022), and evaluate each method on 10 rollouts, covering the distribution of object's initial placement in the dataset. The average success rate (%) and standard deviation across rollouts are reported in Table 1. As shown, VIP-RWR im-

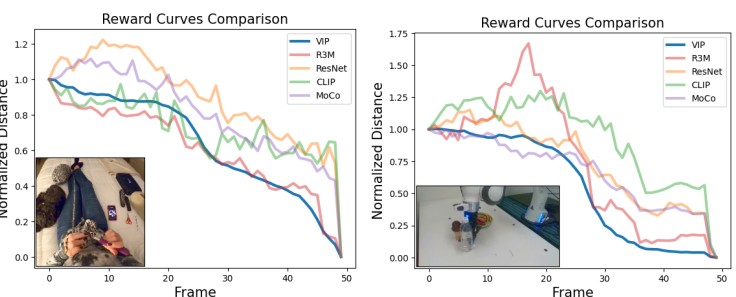

Figure 6: Embedding distance curves on Ego4D (L) and real-robot (R) videos.

proves upon VIP-BC on all tasks and provides substantial benefit on the harder tasks that are multi-stage in nature. In contrast, R3M-RWR, while able to improve R3M-BC on the simpler two tasks involving pushing an object, fails to make any progress on the harder tasks. *In-domain* VIP-based methods fail completely and their actions are almost always pre-empted by the hardware safety

check to prevent robot damage (indicated by $*$), suggesting significant overfitting in training the VIP representation using just the scarce task-specific data $D_{\text{task}}$. The low performance of BC-based methods on the harder `PickPlaceMelon` and `FoldTowel` tasks indicates that in this low-data regime, regardless of what visual representation the policy uses, good reward information is necessary for task success. Altogether, these results corroborate the necessity of pre-training in achieving few-shot offline RL in the real-world and highlight the unique effectiveness of VIP in realizing this goal. Qualitatively, we find VIP-RWR policies acquire intelligent behavior such as robust key action execution and task re-attempt; see App. F.4 and our supplementary video for analysis.

### 5.3 QUALITATIVE ANALYSIS

We hypothesize that VIP learns the most temporally smooth embedding that enables effective zero-shot reward-specification, and present several qualitative experiments investigating this claim. First, in Fig. 6, we visually overlay and compare the embedding distance-to-goal curves for each representation on representative videos from both Ego4D and our real-robot dataset; the curve for every representation is normalized to have initial distance 1 to enable comparison. As shown, VIP has the most visually smooth curves, whereas all other methods exhibit "bumps" (i.e., positive slope at a step) that signal more prevalent presence of local minima in their reward landscapes; in App. G.5, we provide additional embedding curves. Finally, we quantify the total number of "bumps" each representation encounters over both datasets in App. G.6, and find VIP indeed has much fewer bumps.

In addition to the number of bumps, we posit that the *magnitude* of bumps is also a key distinguishing factor among representations. Because the reward (equation 1) is the negative embedding distance-to-goal difference, a positive reward at a step is equivalent to a *negative* slope at

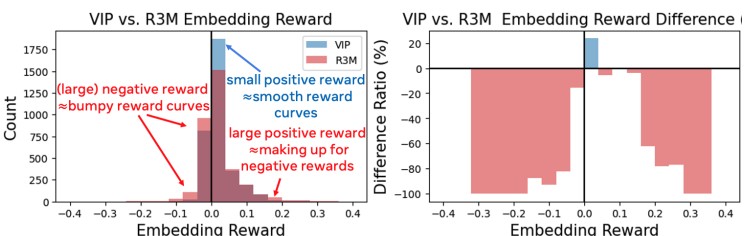

Figure 7: Embedding reward histogram on our real-robot dataset.

that step in the corresponding embedding distance curve. The ideal representation should have a reward histogram that has a tall peak in the first positive bin, indicating that its embedding distance curves consist of mostly *small*, negative slopes that make for smooth curves. In Fig. 7, we compare VIP and R3M representations (comparisons to other baselines are in App. G.7) by overlaying their respective normalized embedding reward histogram computed over the entirety of the real-robot dataset (we include the histograms computed over Ego4D in App. G.8). In addition, we create a bar-plot over the count-difference ratio for each bin, $\frac{|\text{VIP}|-|\text{R3M}|}{|\text{R3M}|}$. We see that VIP has much higher count in the first positive-reward bin ($\approx$+20% more than R3M), fewer negative rewards overall, and much fewer extreme rewards ($\approx$-100% to R3M) in either direction, indicating that on aggregate VIP's reward landscape is much smoother than that of R3M, and this trend holds against all other baselines. These findings confirm our hypothesis that VIP learns the most temporally smooth representation. Surprisingly, despite trained on Ego4D, VIP's smoothness property transfers to the robot domain, suggesting that VIP representation indeed has learned generalizable features that are predictive of goal-directed task progress, and it is this generalization that lies at the heart of VIP's ability to perform zero-shot reward-specification.

### 6 CONCLUSION

We have proposed Value-Implicit Pre-training (VIP), a self-supervised value-based pre-training objective that is highly effective in providing both the visual reward and representation for downstream unseen robotics tasks. VIP is derived from first principles of dual reinforcement learning and admits an appealing connection to an implicit and more powerful formulation of time contrastive learning, which captures long-range temporal dependency and injects local temporal smoothness in the representation to make for effective zero-shot reward specification. Trained entirely on diverse, in-the-wild human videos, VIP demonstrates significant gains over prior state-of-art pre-trained visual representations on an extensive set of policy learning settings. Notably, VIP can enable sample-efficient real-world offline RL with just handful of trajectories. Altogether, we believe that VIP makes an important contribution in both the algorithmic frontier of visual pre-training for RL and practical real-world robot learning.

## ACKNOWLEDGMENT

This work was done while YJM was an intern at Meta AI. We thank Aravind Rajeswaran and members of Meta AI for helpful discussions, Jay Vakil for assistance on the real-robot experiments, and Vincent Moens for assistance on releasing the model.

## REPRODUCIBILITY STATEMENT

We have open-sourced code for using our pre-trained VIP model and training a new VIP model using any custom video dataset at `https://github.com/facebookresearch/vip`; the instruction for model training and inference is included in the `README.md` file in the supplementary file, and the hyperparameters are already configured. A PyTorch-based pseudocode for VIP is also included in Alg. 2.

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

# Part I

# Appendix

## Table of Contents

## A  ADDITIONAL BACKGROUND

### A.1  GOAL-CONDITIONED REINFORCEMENT LEARNING

This section is adapted from Ma et al. (2022b). We consider a goal-conditioned Markov decision process from visual state space: $\mathcal{M} = (O, A, G, r, T, \mu_0, \gamma)$ with state space $O$, action space $A$, reward $r(o, g)$, transition function $o' \sim T(o, a)$, the goal distribution $p(g)$, and the goal-conditioned initial state distribution $\mu_0(o; g)$, and discount factor $\gamma \in (0, 1]$. We assume the state space $O$ and

the goal space $G$ to be defined over RGB images. The objective of goal-conditioned RL is to find a goal-conditioned policy $\pi : O \times G \to \Delta(A)$ that maximizes the discounted cumulative return:

$$J(\pi) := \mathbb{E}_{p(g),\mu_0(o;g),\pi(a_t|s_t,g),T(o_{t+1},|o_t,a_t)} \left[ \sum_{t=0}^{\infty} \gamma^t r(o_t; g) \right] \tag{8}$$

The *goal-conditioned* state-action occupancy distribution $d^\pi(o, a; g) : O \times A \times G \to [0, 1]$ of $\pi$ is

$$d^\pi(o, a; g) := (1 - \gamma) \sum_{t=0}^{\infty} \gamma^t \Pr(o_t = o, a_t = a \mid o_0 \sim \mu_0(o; g), a_t \sim \pi(o_t; g), o_{t+1} \sim T(o_t, a_t)) \tag{9}$$

which captures the goal-conditioned visitation frequency of state-action pairs for policy $\pi$. The state-occupancy distribution then marginalizes over actions: $d^\pi(o; g) = \sum_a d^\pi(o, a; g)$. Then, it follows that $\pi(a \mid o, g) = \frac{d^\pi(o,a;g)}{d^\pi(o;g)}$. A state-action occupancy distribution must satisfy the *Bellman flow constraint* in order for it to be an occupancy distribution for some stationary policy $\pi$:

$$\sum_a d(o, a; g) = (1 - \gamma)\mu_0(o; g) + \gamma \sum_{\tilde{o},\tilde{a}} T(s \mid \tilde{o}, \tilde{a})d(\tilde{o}, \tilde{a}; g), \qquad \forall o \in O, g \in G \tag{10}$$

We write $d^\pi(o, g) = p(g)d^\pi(o; g)$ as the joint goal-state density induced by $p(g)$ and the policy $\pi$. Finally, given $d^\pi$, we can express the objective function equation 8 as $J(\pi) = \frac{1}{1-\gamma}\mathbb{E}_{(o,g)\sim d^\pi(o,g)}[r(o, g)]$.

## A.2 INFONCE & TIME CONTRASTIVE LEARNING.

As VIP can be understood as a implicit and smooth time contrastive learning objective, we provide additional background on the InfoNCE Oord et al. (2018) and time contrastive learning (TCN) (Sermanet et al., 2018) objective to aid comparison in Section 4.2.

InfoNCE is an unsupervised contrastive learning objective built on the noise contrastive estimation (Gutmann & Hyvärinen, 2010) principle. In particular, given an "anchor" datum $x$ (otherwise known as context), and distribution of positives $x_{\text{pos}}$ and negatives $x_{\text{neg}}$, the InfoNCE objective optimizes

$$\min_\phi \mathbb{E}_{x_{\text{pos}}} \left[ -\log \frac{\mathcal{S}_\phi(x, x_{\text{pos}})}{\mathbb{E}_{x_{\text{neg}}} \mathcal{S}_\phi(x, x_{\text{neg}})} \right], \tag{11}$$

where $\mathbb{E}_{x_{\text{neg}}}$ is often approximated with a fixed number of negatives in practice.

It is shown in Oord et al. (2018) that optimizing equation 11 is maximizing a lower bound on the mutual information $\mathcal{I}(x, x_{\text{pos}})$, where, with slight abuse of notation, $x$ and $x_{\text{pos}}$ are interpreted as random variables.

TCN is a contrastive learning objective that learns a representation that in timeseries data (e.g., video trajectories). The original work (Sermanet et al., 2018) considers multi-view videos and perform contrastive learning over frames in separate videos; in this work, we consider the single-view variant. At a high level, TCN attracts representations of frames that are temporally close, while pushing apart those of frames that are farther apart in time. More precisely, given three frames sampled from a video sequence $(o_{t_1}, o_{t_2}, o_{t_3})$, where $t_1 < t_2 < t_3$, TCN would attract the representations of $o_{t_1}$ and $o_{t_2}$ and repel the representation of $o_{t_3}$ from $o_{t_1}$. This idea can be formally expressed via the following objective:

$$\min_\phi \mathbb{E}_{(o_{t_1}, o_{t_2 > t_1}) \sim D} \left[ -\log \frac{e^{-\|\phi(o_{t_1}) - \phi(o_{t_2})\|_2}}{\mathbb{E}_{o_{t_3}|t_3 > t_2 \sim D} \left[ \exp\left( -\|\phi(o_{t_1}) - \phi(o_{t_3})\|_2 \right) \right]} \right] \tag{12}$$

Given a "positive" window of $K$ steps and a uniform distribution among valid positive samples, we can write equation 12 as

$$\min_\phi \frac{1}{K} \sum_{k=1}^{K} \mathbb{E}_{(o_{t_1}, o_{t_1+k}) \sim D} \left[ -\log \frac{\mathcal{S}_\phi(o_{t_1}; o_{t_1+k})}{\mathbb{E}_{o_{t_3}|t_3 > t_1+k \sim D} \left[ \mathcal{S}_\phi(o_{t_1}; o_{t_3}) \right]} \right], \tag{13}$$

in which each term inside the expectation is a standalone InfoNCE objective tailored to observation sequence data.

## B EXTENDED RELATED WORK

**Perceptual Reward Learning from Human Videos.** Human videos provide a rich natural source of reward and representation learning for robotic learning. Most prior works exploit the idea of learning an invariant representation between human and robot domains to transfer the demonstrated skills (Sermanet et al., 2016; 2018; Schmeckpeper et al., 2020; Chen et al., 2021; Xiong et al., 2021; Zakka et al., 2022; Bahl et al., 2022). However, training these representations require task-specific human *demonstration* videos paired with robot videos solving the same task, and cannot leverage the large amount of "in-the-wild" human videos readily available. As such, these methods require robot data for training, and learn rewards that are task-specific and do not generalize beyond the tasks they are trained on. In contrast, VIP do not make any assumption on the quality or the task-specificity of human videos and instead pre-trains an (implicit) value function that aims to capture task-agnostic goal-oriented progress, which can generalize to completely unseen robot domains and tasks.

**Goal-Conditioned RL as Representation Learning.** Our pre-training method is also related to the idea of treating goal-conditioned RL as representation learning. Chebotar et al. (2021) shows that a goal-conditioned Q-function trained with offline in-domain multi-task robot data learns an useful visual representation that can accelerate learning for a new downstream task in the same domain. Eysenbach et al. (2022) shows that goal-conditioned Q-learning with a particular choice of reward function can be understood as performing contrastive learning. In contrast, our theory introduces a new implicit time contrastive learning, and states that for *any* choice of reward function, the dual formulation of a regularized offline GCRL objective can be cast as implicit time contrast. This conceptual bridge also explains why VIP's learned embedding distance is temporally smooth and can be used as an universal reward mechanism. Finally, whereas these two works are limited to training on in-domain data with robot action labels, VIP is able to leverage diverse out-of-domain human data for visual representation pre-training, overcoming the inherent limitation of robot data scarcity for in-domain training.

Our work is also closely related to Ma et al. (2022b), which first introduced the dual offline GCRL objective based on Fenchel duality (Rockafellar, 1970; Nachum & Dai, 2020; Ma et al., 2022a). Whereas Ma et al. (2022b) assumes access to the true state information and focuses on the offline GCRL setting using in-domain offline data with robot action labels, we extend the dual objective to enable out-of-domain, action-free pre-training from human videos. Our particular dual objective also admits a novel implicit time contrastive learning interpretation, which simplifies VIP's practical implementation by letting the value function be implicitly defined instead of a deep neural network as in Ma et al. (2022b).

## C TECHNICAL DERIVATIONS AND PROOFS

### C.1 PROOF OF PROPOSITION 4.1

We first reproduce Proposition 4.1 for ease of reference:

**Proposition C.1.** *Under assumption of deterministic transition dynamics, the dual optimization problem of*

$$\max_{\pi_H,\phi} \mathbb{E}_{\pi^H}\left[\sum_t \gamma^t r(o;g)\right] - D_{\mathrm{KL}}(d^{\pi_H}(o,a^H;g)\|d^D(o,\tilde{a}^H;g)), \tag{14}$$

*is*

$$\max_\phi \min_V \mathbb{E}_{p(g)}\left[(1-\gamma)\mathbb{E}_{\mu_0(o;g)}[V(\phi(o);\phi(g))] + \log\mathbb{E}_{D(o,o';g)}\left[\exp\left(r(o,g)+\gamma V(\phi(o');\phi(g))-V(\phi(o),\phi(g))\right)\right]\right], \tag{15}$$

*where $\mu_0(o;g)$ is the goal-conditioned initial observation distribution, and $D(o,o';g)$ is the goal-conditioned distribution of two consecutive observations in dataset $D$.*

*Proof.* We begin by rewriting equation 14 as an optimization problem over valid state-occupancy distributions. To this end, we have[1]

$$\max_{\phi} \max_{d(\phi(o),a;\phi(g)) \geq 0} \mathbb{E}_{d(\phi(o),\phi(g))}\left[r(o;g)\right] - D_{\mathrm{KL}}(d(\phi(o),a;\phi(g))\|d^D(\phi(o),\tilde{a};\phi(g)))$$

(P)   s.t.   $$\sum_a d(\phi(o),a;\phi(g)) = (1-\gamma)\mu_0(o;g) + \gamma \sum_{\tilde{o},\tilde{a}} T(o \mid \tilde{o},\tilde{a})d(\phi(\tilde{o}),\tilde{a};\phi(g)), \forall o \in O, g \in G$$

(16)

Fixing a choice of $\phi$, the inner optimization problem operates over a $\phi$-induced state and goal space, giving us equation 16. Then, applying Proposition 4.2 of Ma et al. (2022b) to the inner optimization problem, we immediately obtain

$$\max_{\phi} \min_V \mathbb{E}_{p(g)}\big[(1-\gamma)\mathbb{E}_{\mu_0(o;g)}[V(\phi(o);\phi(g))]$$

(D)   $$+ \log \mathbb{E}_{d^D(\phi(o),a;\phi(g))}\big[\exp\big(r(o,g) + \gamma\mathbb{E}_{T(o'|o,a)}[V(\phi(o');\phi(g))] - V(\phi(o),\phi(g))\big)\big]\big]$$

(17)

Now, given our assumption that the transition dynamics is deterministic, we can replace the inner expectation $\mathbb{E}_{T(o'|o,a)}$ with just the observed sample in the offline dataset and obtain:

$$\max_{\phi} \min_V \mathbb{E}_{p(g)}\big[(1-\gamma)\mathbb{E}_{\mu_0(o;g)}[V(\phi(o);\phi(g))]$$

$$+ \log \mathbb{E}_{d^D(\phi(o),\phi(o');\phi(g))}\big[\exp\big(r(o,g) + \gamma V(\phi(o');\phi(g)) - V(\phi(o),\phi(g))\big)\big]\big]$$

(18)

Finally, sampling embedded states from $d^D(\phi(o),\phi(o');\phi(g))$ is equivalent to sampling from $D(o,o';g)$, assuming there is no embedding collision (i.e., $\phi(o) \neq \phi(o'), \forall o \neq o'$), which can be satisfied by simply augmenting any $\phi$ by concatenating the input to the end. Then, we have our desired expression:

$$\max_{\phi} \min_V \mathbb{E}_{p(g)}\big[(1-\gamma)\mathbb{E}_{\mu_0(o;g)}[V(\phi(o);\phi(g))] + \log \mathbb{E}_{D(o,o';g)}\big[\exp\big(r(o,g) + \gamma V(\phi(o');\phi(g)) - V(\phi(o),\phi(g))\big)\big]\big]$$

(19)

$\square$

## C.2   VIP Implicit Time Contrast Learning Derivation

This section provides all intermediate steps to go from equation 4 to equation 5. First, we have

$$\min_{\phi} \mathbb{E}_{p(g)}\left[(1-\gamma)\mathbb{E}_{\mu_0(o;g)}[-V^*(\phi(o);\phi(g))] + \log \mathbb{E}_{D(o,o';g)}\left[\exp\left(\tilde{\delta}_g(o) + \gamma V(\phi(o');\phi(g)) - V(\phi(o),\phi(g))\right)\right]^{-1}\right].$$

(20)

We can equivalently write this objective as

$$\min_{\phi} \mathbb{E}_{p(g)}\left[(1-\gamma)\mathbb{E}_{\mu_0(o;g)}[-\log e^{V^*(\phi(o);\phi(g))}] + \log \mathbb{E}_{D(o,o';g)}\left[\exp\left(\tilde{\delta}_g(o) + \gamma V(\phi(o');\phi(g)) - V(\phi(o),\phi(g))\right)\right]^{-1}\right].$$

(21)

Then,

$$\min_{\phi} \mathbb{E}_{p(g)}\left[(1-\gamma)\mathbb{E}_{\mu_0(o;g)}\left[-\log e^{V^*(\phi(o);\phi(g))} - \log \mathbb{E}_{D(o,o';g)}\left[\exp\left(\tilde{\delta}_g(o) + \gamma V(\phi(o');\phi(g)) - V(\phi(o),\phi(g))\right)\right]^{\frac{-1}{1-\gamma}}\right]\right]$$

$$= \min_{\phi}(1-\gamma)\mathbb{E}_{p(g),\mu_0(o;g)}\left[\log \frac{e^{-V^*(\phi(o);\phi(g))}}{\mathbb{E}_{D(o,o';g)}\left[\exp\left(\tilde{\delta}_g(o) + \gamma V(\phi(o');\phi(g)) - V(\phi(o),\phi(g))\right)\right]^{\frac{-1}{1-\gamma}}}\right].$$

(22)

This is equation 5 in the main text.

## C.3   VIP Implicit Repulsion

In this section, we formalize the implicit repulsion property of VIP objective (equation 5); in particular, we prove that under certain assumptions, it always holds for optimal paths.

---

[1]We omit the human action superscript $H$ in this derivation.

**Proposition C.2.** *Suppose* $V^*(s; g) := -\|\phi(s) - \phi(g)\|_2$ *for some* $\phi$, *under the assumption of deterministic dynamics (as in Proposition 4.1), for any pair of consecutive states reached by the optimal policy,* $(s_t, s_{t+1}) \sim \pi^*$, *we have that*

$$\|\phi(s_t) - \phi(g)\|_2 > \|\phi(s_{t+1}) - \phi(g)\|_2 , \tag{23}$$

*Proof.* First, we note that

$$V^*(s; g) = \max_a Q^*(s, a; g) \tag{24}$$

A proof can be found in Section 1.1.3 of Agarwal et al. (2019). Then, due to the Bellman optimality equation, we have that

$$Q^*(s, a; g) = r(s, g) + \gamma \mathbb{E}_{s' \sim T(s,a)} \max_{a'} Q^*(s', a'; g) \tag{25}$$

Given that the dynamics is deterministic and equation 24, we have that

$$Q^*(s, a; g) = r(s, g) + \gamma V^*(s'; g) \tag{26}$$

Now, for $(s_t, a_t, s_{t+1}) \sim \pi^*$, this further simplifies to

$$V^*(s_t; g) = r(s_t, g) + \gamma V^*(s_{t+1}; g) \tag{27}$$

Note that since $V^*$ is also the optimal value function, given that $r(s_t, g) = \mathbb{I}(s_t = g) - 1$, $V^*(s_t; g)$ is the *negative* discounted distance of the shortest path between $s_t$ ans $g$. In particular, since $V^*(g; g) = 0$ by construction, we have that $V^*(s_t; g) = -\sum_{k=0}^K \gamma^k$ (this also clearly satisfies equation 27), where the shortest path (i.e., the path $\pi^*$ takes) between $s_t$ and $g$ are $K$ steps long. Now, giving that we assume $V^*(s_t; g)$ can be expressed as $-\|\phi(s_t) - \phi(g)\|_2$ for some $\phi$, it immediately follows that

$$\|\phi(s_t) - \phi(g)\|_2 > \|\phi(s_{t+1}) - \phi(g)\|_2 , \quad \forall (s_t, s_{t+1}) \sim \pi^* \tag{28}$$

$\square$

The implication of this result is that at least along the trajectories generated by the optimal policy, the representation will have monotonically decreasing and well-behaved embedding distances to the goal. Now, since in practice, VIP is trained on goal-directed (human video) trajectories, which are near-optimal for goal-reaching, we expect this smoothness result to be informative about VIP's embedding practical behavior and help formalize out intuition about the mechanism of implicit time contrastive learning. As confirmed by our qualitative study in Section 5.3, We highlight that VIP's embedding is indeed much smoother than other baselines along test trajectories on both Ego4D and on our real-robot dataset. This smoothness along optimal paths makes it easier for the downstream control optimizer to discover these paths, conferring VIP representation effective zero-shot reward-specification capability that is not attained by any other comparison.

# D VIP TRAINING DETAILS

## D.1 DATASET PROCESSING AND SAMPLING

We use the exact same pre-processed Ego4D dataset as in R3M, in which long raw videos are first processed into shorter videos consisting of 10-150 frames each. In total, there are approximately 72000 clips and 4.3 million frames in the dataset. Within a sampled batch, we first sample a set of videos, and then sample a sub-trajectory from each video (Step 3 in Algorithm 1). In this formulation, each sub-trajectory is treated as a video segment from the algorithm's perspective; this can viewed as a variant of trajectory data augmentation. As in R3M, we apply random crop at a video level within a batch, so all frames from the same video sub-trajectory are cropped the same way. Then, each raw observation is resized and center-cropped to have shape $224 \times 224 \times 3$ before passed into the visual encoder. Finally, as in standard contrastive learning and R3M, for each sampled sub-trajectory $\{o_t^i, ..., o_k^i, o_{k+1}^i, ..., o_T^i\}$, we also sample additional 3 negative samples $(\tilde{o}_j, \tilde{o}_{j+1})$ from separate video sequences to be included in the log-sum-exp term in $\mathcal{L}(\phi)$.

## D.2 VIP HYPERPARAMETERS

Hyperparameters used can be found in Table 2.

Table 2: VIP Architecture & Hyperparameters.

|  | Name | Value |
|---|---|---|
| Architecture | Visual Backbone | ResNet50 (He et al., 2016) |
|  | FC Layer Output Dim | 1024 |
| Hyperparameters | Optimizer | Adam (Kingma & Ba, 2014) |
|  | Learning rate | 0.0001 |
|  | $L_1$ weight penalty | 0.001 |
|  | $L_1$ weight penalty | 0.001 |
|  | Mini-batch size | 32 |
|  | Discount factor $\gamma$ | 0.98 |

### D.3 VIP PYTORCH PSEUDOCODE

In this section, we present a pseudocode of VIP written in PyTorch (Paszke et al., 2019), Algorithm 2. As shown, the main training loop can be as short as 10 lines of code.

**Algorithm 2** VIP PyTorch Pseudocode

```
# D: offline dataset
# phi: vision architecture

# training loop
for (o_0, o_t1,o_t2, g) in D:
    phi_g = phi(o_g)
    V_0 = - torch.linalg.norm(phi(o_0), phi_g)
    V_t1 = - torch.linalg.norm(phi(o_t1), phi_g)
    V_t2 = - torch.linalg.norm(phi(o_t2), phi_g)
    VIP_loss = (1-gamma)*-V_0.mean() + torch.logsumexp(V_t1+1-gamma*V_t2)
    optimizer.zero_grad()
    VIP_loss.backward()
    optimizer.step()
```

## E  SIMULATION EXPERIMENT DETAILS.

### E.1  FRANKAKITCHEN TASK DESCRIPTIONS

In this section, we describe the FrankaKitchen suite for our simulation experiments. We use 12 tasks from the `v0.1` version[2] of the environment.

We use the environment default initial state as the initial state and frame for all tasks in the Hard setting. In the Easy setting, we use the 20th frame of a demonstration trajectory and its corresponding environment state as the initial frame and state. The goal frame for both settings is chosen to be the last frame of the same demonstration trajectory. The initial frames and goal frame for all 12 tasks and 3 camera views are illustrated in Figure 8-9. In the Easy setting, the horizon for all tasks is 50 steps; in the Hard setting, the horizon is 100 steps. Note that using the 20th frame as the initial state is a crude way for initializing the robot, and for some tasks, this initialization makes the task substantially easier, whereas for others, the task is still considerably difficult. Furthermore, some tasks become naturally more difficult depending on camera viewpoints. For these reasons, it is worth noting that our experiment's emphasis is on the *aggregate* behavior of pre-trained representations, instead of trying to solve any particular task as well as possible.

### E.2  IN-DOMAIN REPRESENTATION PROBING

In this section, we describe the experiment we performed to generate the in-domain VIP vs. TCN comparison in Figure 2. We fit VIP and TCN representations using 100 demonstrations from the

---

[2]https://github.com/vikashplus/mj_envs/tree/v0.1real/mj_envs/envs/relay_kitchen

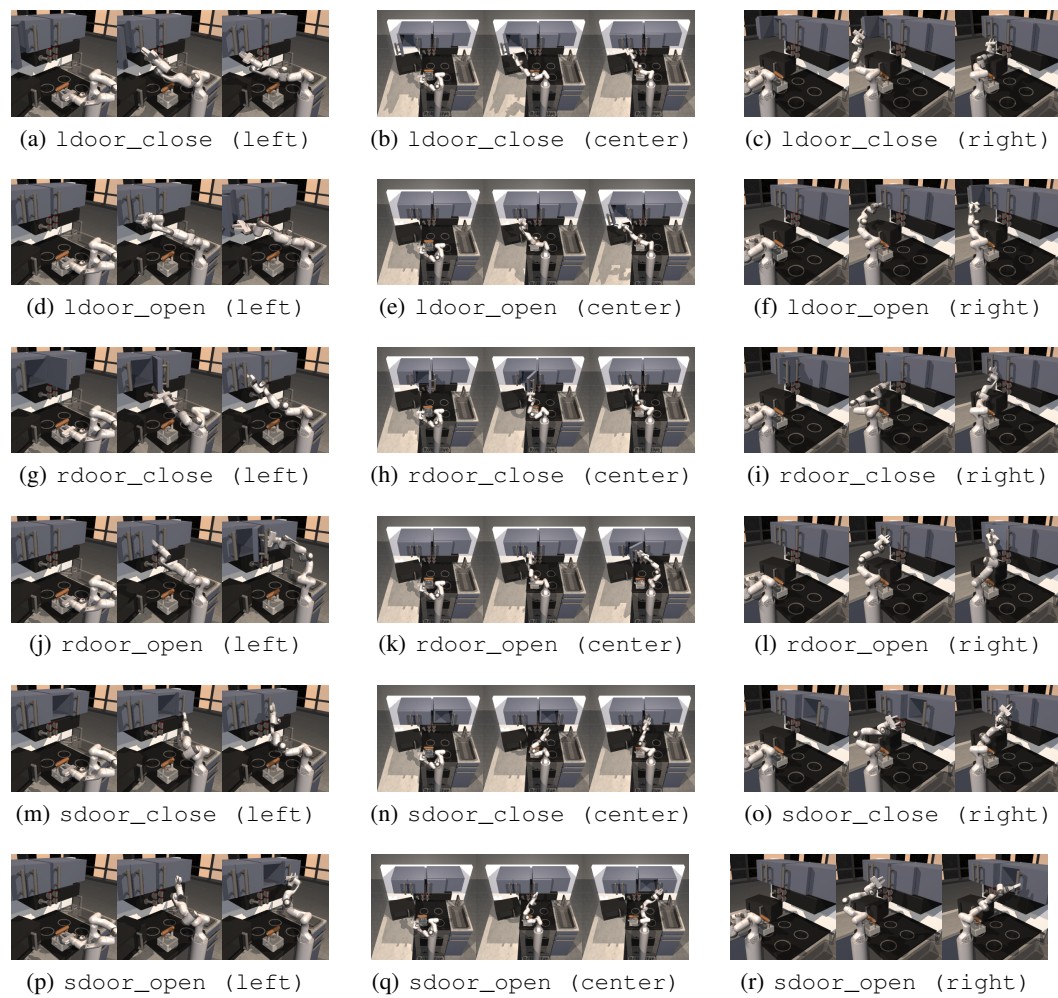

Figure 8: Initial frame (Easy), initial frame (Hard), and goal frame for all 12 tasks and 3 camera views in our FrankaKitchen suite.

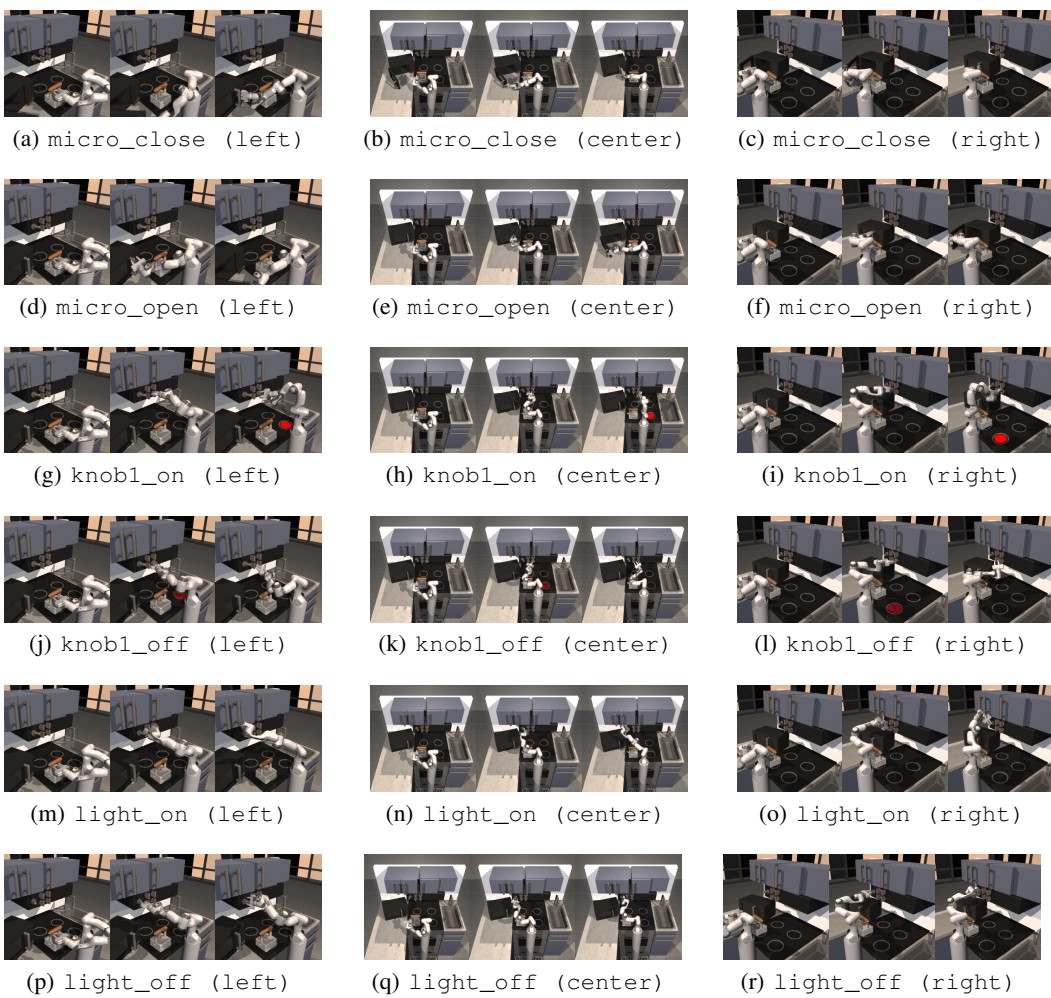

Figure 9: Initial frame (Easy), initial frame (Hard), and goal frame for all 12 tasks and 3 camera views in our FrankaKitchen suite.

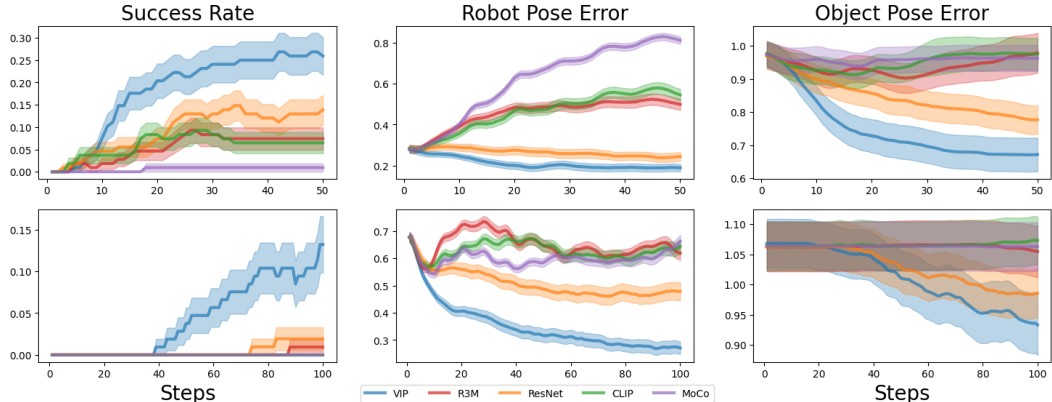

Figure 10: Trajectory optimization results with pose errors.

FrankaKitchen `sdoor_open` task (center view). For TCN, we use R3M's implementation of the TCN loss without any modification; this also allows our findings in Figure 2 to extend to the main experiment section. The visual architecture is ResNet34, and the output dimension is 2, which enables us to directly visualize the learned embedding. Different from the out-of-domain version of VIP, we also do not perform weight penalty, trajectory-level random cropping data augmentation, or additional negative sampling. Besides these choices, we use the same hyperparameters as in Table 2 and train for 2000 batches.

### E.3 TRAJECTORY OPTIMIZATION

We use a publicly available implementation of MPPI[3], and make no modification to the algorithm or the default hyperparameters. In particular, the planning horizon is 12 and 32 sequences of actions are proposed per action step. Because the embedding reward (equation 1) is the goal-embedding distance difference, the score (i.e., sum of per-transition reward) of a proposed sequence of actions is equivalent to the negative embedding distance (i.e., $S_\phi(\phi(o_T); \phi(g))$) at the last observation.

#### E.3.1 ROBOT AND OBJECT POSE ERROR ANALYSIS

In this section, we visualize the per-step robot and object pose $L_2$ error with respect to the goal-image poses. We report the non-cumulative curves (on the success rate as well) for more informative analysis.

### E.4 REINFORCEMENT LEARNING

We use a publicly available implementation of NPG[4], and make no modification to the algorithm or the default hyperparameters. In the Easy (resp. Hard) setting, we train the policy until 500000 (resp. 1M) real environment steps are taken. For evaluation, we report the cumulative maximum success rate on 50 test rollouts from each task configuration (50*108=5400 total rollouts) every 10000 step.

## F REAL-WORLD ROBOT EXPERIMENT DETAILS

### F.1 TASK DESCRIPTIONS

The robot learning environment is illustrated in Figure 11; a RealSense camera is mounted on the right edge of the table, and we only use the RGB image stream without depth information for data collection and policy learning.

We collect offline data $D_{\text{task}}$ for each task via kinesthetic playback, and the object initial placement is randomized for each trajectory. On the simplest `CloseDrawer` task, we combine 10 expert demonstrations with 20 sub-optimal failure trajectories to increase learning difficulty. For the other

---

[3]https://github.com/aravindr93/trajopt/blob/master/trajopt/algos/mppi.py
[4]https://github.com/aravindr93/mjrl/blob/master/mjrl/algos/npg_cg.py

Table 3: Real-world robotics tasks descriptions.

| Environment | Object Type | Dataset | Success Criterion |
|---|---|---|---|
| CloseDrawer | Articulated Object | 10 demos + 20 failures | the drawer is closed enough that the spring loads. |
| PushBottle | Transparent Object | 20 demonstrations | the bottle is parallel to the goal line set by the icecream cone. |
| PlaceMelon | Soft Object | 20 demonstrations | the watermelon toy is fully placed in the plate. |
| FoldTowel | Deformable Object | 20 demonstrations | the bottom half of the towel is cleanly covered by the top half. |

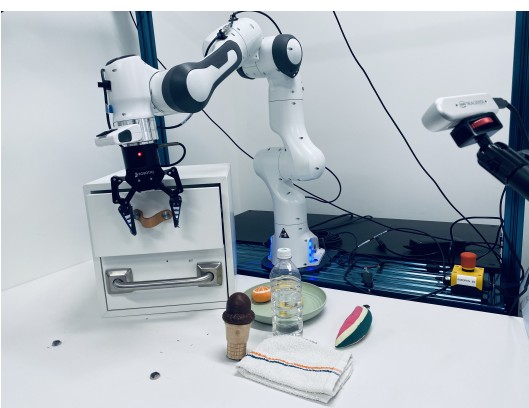

Figure 11: Real-robot setup.

three tasks, we collect 20 expert demonstrations, which we found are difficult enough for learning good policies. Each demonstration is 50-step long collected at 25Hz. The initial state for the robot is fixed for each demonstration and test rollout, but the object initial position is randomized. The task success is determined based on a visual criterion that we manually check for each test rollout. The full task breakdown is described in Table 3.

Each task is specified via a set of goal images that are chosen to be the last frame of all demonstrations for the task. Hence, the goal embedding used to compute the embedding reward (equation 1( for each task is the average over the embeddings of all goal frames.

The tasks (in their initial positions) using a separate high-resolution phone camera are visualized in Figure 12. Sample demonstrations in the robot camera view are visualized in Figure 13.

### F.2 TRAINING AND EVALUATION DETAILS

The policy network is implemented as a 2-layer MLP with hidden sizes [256, 256]. As in R3M's real-world robot experiment setup, the policy takes in concatenated visual embedding of current observation and robot's proprioceptive state and outputs robot action. The policy is trained with a learning rate of 0.001, and a batch size of 32 for 20000 steps.

For RWR's temperature scale, we use $\tau = 0.1$ for all tasks, except CloseDrawer where we find $\tau = 1$ more effective for both VIP and R3M.

For policy evaluation, we use 10 test rollouts with objects randomly initialized to reflect the object distribution in the expert demonstrations. The rollout horizon is 100 steps.

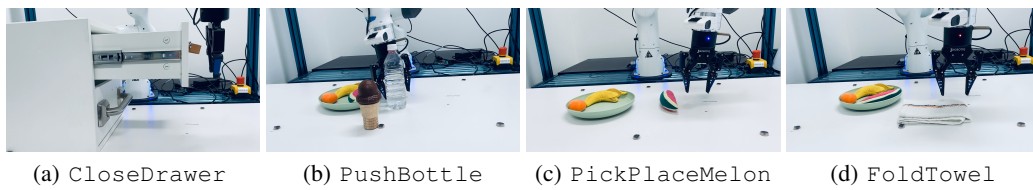

(a) CloseDrawer  (b) PushBottle  (c) PickPlaceMelon  (d) FoldTowel

Figure 12: Side-view of real-robot tasks using a high-resolution smartphone camera.

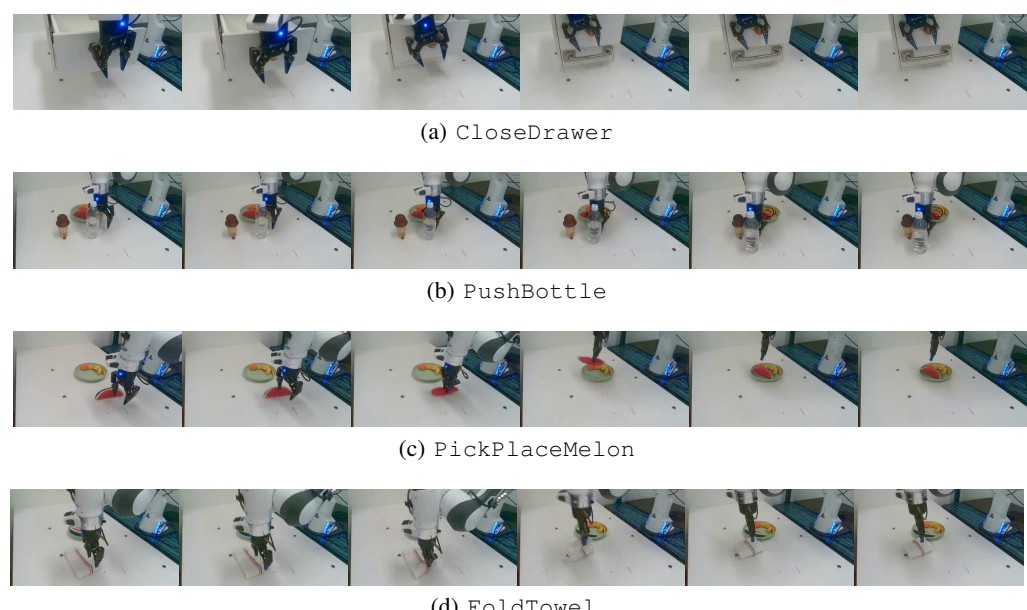

(a) `CloseDrawer`

(b) `PushBottle`

(c) `PickPlaceMelon`

(d) `FoldTowel`

Figure 13: Real-robot task demonstrations (every 10th frame) in robot camera view. The first and last frames in each row are representative of initial and final goal observaions for the respective task.

### F.3    ADDITIONAL ANALYSIS & CONTEXT

**Offline RL vs. imitation learning for real-world robot learning.** Offline RL, though known as the data-driven paradigm of RL (Levine et al., 2020), is not necessarily data *efficient* (Agarwal et al., 2021), requiring hundreds of thousands of samples even in low-dimensional simulated tasks, and requires a dense reward to operate most effectively (Mandlekar et al., 2021; Yu et al., 2022). Furthermore, offline RL algorithms are significantly more difficult to implement and tune compared to BC (Kumar et al., 2021; Zhang & Jiang, 2021). As such, the dominant paradigm of real-world robot learning is still learning from demonstrations (Jang et al., 2022; Mandlekar et al., 2018; Ebert et al., 2021). With the advent of VIP-RWR, offline RL may finally be a practical approach for real-world robot learning at scale.

**Performance of R3M-BC.** Our R3M-BC, though able to solve some of the simpler tasks, appears to perform relatively worse than the original R3M-BC in Nair et al. (2022) on their real-world tasks. To account for this discrepancy, we note that our real-world experiment uses different software-hardware stacks and tasks from the original R3M real-world experiments, so the results are not directly comparable. For instance, camera placement, an important variable for real-world robot learning, is chosen differently in our experiment and that of R3M; in R3M, a different camera angle is selected for each task, whereas in our setup, the same camera view is used for all tasks. Furthermore, we emphasize that our focus is not the absolute performance of R3M-BC, but rather the relative improvement R3M-RWR provides on top of R3M-BC.

### F.4    QUALITATIVE ANALYSIS

In this section, we study several interesting policy behaviors VIP-RWR acquire. Policy videos are included in our supplementary video.

**Robust key action execution.** VIP-RWR is able to execute key actions more robustly than the baselines; this suggests that its reward information helps it identify necessary actions. For example, as shown in Figure 14, on the `PickPlaceMelon` task, failed VIP-RWR rollouts at least have the gripper grasp onto the watermelon, whereas for other baselines, the failed rollouts do not have the watermelon between the gripper and often incorrectly push the watermelon to touch the plate's outer edge, preventing pick-and-place behavior from being executed.

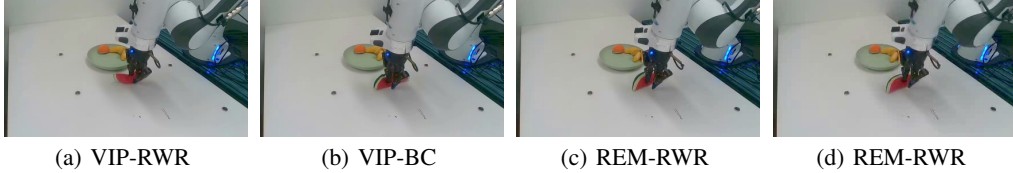

| (a) VIP-RWR | (b) VIP-BC | (c) REM-RWR | (d) REM-RWR |

Figure 14: Comparison of failure trajectories on `PickPlaceMelon`. VIP-RWR is still able to reach the critical state of gripping watermelon, whereas baselines fail.

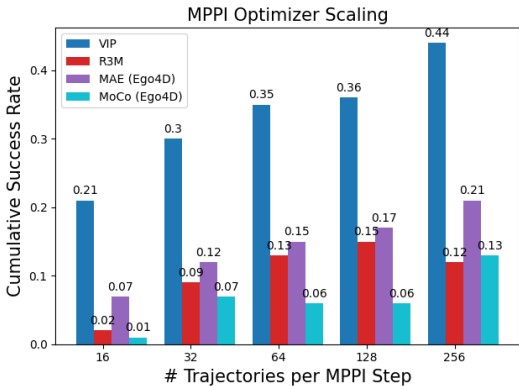

Figure 15: VIP vs. Alternative Pre-Training Algorithms on Ego4D.

**Task re-attempt.** We observe that VIP-RWR often learns more robust policies that are able to perform recovery actions when the task is not solved on the first attempt. For instance, in both `CloseDrawer` and `FoldTowel`, there are trials where VIP-RWR fails to close the drawer all the way or pick up the towel edge right away; in either case, VIP-RWR is able to re-attempt and solves the task (see our supplementary video). This is a known advantage of offline RL over BC (Kumar et al., 2022; Levine et al., 2020); however, we only observe this behavior in VIP-RWR and not R3M-RWR, indicating that this advantage of offline RL is only realized when the reward information is sufficiently informative.

# G  ADDITIONAL RESULTS

## G.1  COMPARISON TO MAE AND MOCO TRAINED ON EGO4D

In this section, we compare to two additional representations trained on the same pre-training dataset of Ego4D. We compare them to VIP and R3M in the trajectory optimization setting and evaluate their performance as the optimization budget increases in the spirit of Figure 5. The result are shown in Figure 15. Consistent with the findings in the main text, the pre-training dataset is not the source of VIP's empirical gains; all prior state-of-art pre-training methods struggle as zero-shot reward functions. Notably, MAE uses a vision transformer (Dosovitskiy et al., 2020) architecture backbone and exhibits an improving trend in performance similar to VIP; however, MAE's absolute performance is still far inferior to VIP.

## G.2  VALUE-BASED PRE-TRAINING ABLATION: LEAST-SQUARE TEMPORAL-DIFFERENCE

While VIP is the first value-based pre-training approach and significantly outperforms all existing methods, we show that this effectiveness is also unique to VIP and not to training a value function. To this end, we show that a simpler value-based baseline does not perform as well. In particular, we consider Least-Square Temporal-Difference policy *evaluation* (**LSTD**) (Bradtke & Barto, 1996;

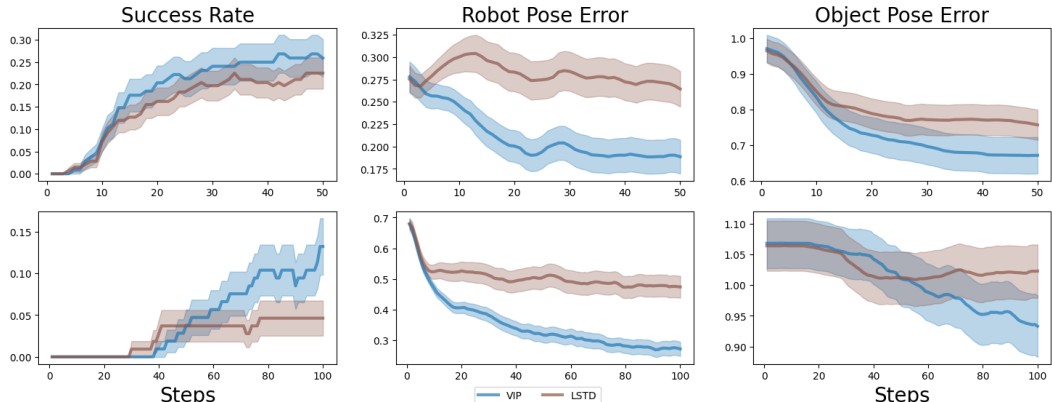

Figure 16: VIP vs. LSTD Trajectory Optimization Comparison.

Table 4: Visual Imitation Learning Results.

|  | Self-Supervised | | | Supervised | | |
|  | VIP (E) | LSTD (E) | R3M-Lang (E) | MOCO (I) | R3M (E) | ResNet50 (I) | CLIP (Internet) |
|---|---|---|---|---|---|---|---|
| Success Rate | **53.6** | 51.5 | 51.2 | 45.0 | **55.9** | 41.8 | 44.3 |

Sutton & Barto, 2018) to assess the importance of the choice of value-training objective:

$$\min_{\phi} \mathbb{E}_{(o,o',g) \sim D} \left[ \left( \tilde{\delta}_g(o) + \gamma V(\phi(o'); \phi(g)) - V(\phi(s), \phi(g)) \right)^2 \right], \tag{29}$$

in which we also parameterize $V$ as the negative $L_2$ embedding distance as in VIP. Given that human videos are reasonably goal-directed, the value of the human behavioral policy computed via LSTD should be a decent choice of reward; however, LSTD does not capture the long-range dependency of initial to goal frames (first term in equation 3), nor can it obtain a value function that outperforms that of the behavioral policy. We train LSTD using the exact same setup as in VIP, differing in only the training objective, and compare it against VIP in our trajectory optimization settings.

As shown in Fig. 16, interestingly, LSTD already works better than all prior baselines in the Easy setting, indicating that value-based pre-training is indeed favorable for reward-specification. However, its inability to capture long range temporal dependency as in VIP (the first term in VIP's objective) makes it far less effective on the Hard setting, which require extended smoothness in the reward landscape to solve given the distance between the initial observation and the goal. These results show that VIP's superior reward specification comes precisely from its ability to capture both long-range temporal dependencies and local temporal smoothness, two innate properties of its dual value objective and the associated implicit time contrastive learning interpretation. To corroborate these findings, we have also included LSTD in our qualitative reward curve and histogram analysis in App. G.5, G.7, and G.8 and finds that VIP generates much smoother embedding than LSTD.

### G.3 VISUAL IMITATION LEARNING

One alternative hypothesis to VIP's smoother embedding for its superior reward-specification capability is that it learns a better visual representation, which then naturally enables a better visual reward function. To investigate this hypothesis, we compare representations' capability as a pure visual encoder in a visual imitation learning setup. We follow the training and evaluation protocol of (Nair et al., 2022) and consider 12 tasks combined from FrankaKitchen, MetaWorld (Yu et al., 2020), and Adroit (Rajeswaran et al., 2017), 3 camera views for each task, and 3 demonstration dataset sizes, and report the aggregate average maximum success rate achieved during training. **R3M-Lang** is the publicly released R3M variant without supervised language training. The average success rates over all tasks are shown in Table 4; the letter inside () stands for the pre-training dataset with $E$ referring to Ego4D and $I$ Imagenet.

These results suggest that with current pre-training methods, the performance on visual imitation learning may largely be a function of the pre-training dataset, as all methods trained on Ego4D, even our simple baseline LSTD, performs comparably and are much better than the next best baseline not trained on Ego4D. Conversely, this result also suggests that despite not being designed for this purely supervised learning setting, value-based approaches constitute a strong baseline, and VIP is in fact currently the state-of-art for self-supervised methods. While these results highlight that VIP is effective even as a pure visual encoder, a necessary requirement for joint effectiveness for visual reward and representation, it fails to explain why VIP is far superior to R3M in reward-based policy learning. As such, we conclude that studying representations' capability as a pure visual encoder may not be sufficient for distinguishing representations that can additionally perform zero-shot reward-specification.

### G.4    EMBEDDING AND TRUE REWARDS CORRELATION

In this section, we create scatterplots of embedding reward vs. true reward on the trajectories MPPI have generated to assess whether the embedding reward is correlated with the ground-truth dense reward. More specifically, for each transition in the MPPI trajectories in Figure 4, we plot its reward under the representation that was used to compute the reward for MPPI versus the true human-crafted reward computed using ground-truth state information. The dense reward in FrankaKitchen tasks is a weighted sum of (1) the negative object pose error, (2) the negative robot pose error, (3) bonus for robot approaching the object, and (4) bonus for object pose error being small. This dense reward is highly tuned and captures human intuition for how these tasks ought to be best solved. As such, high correlation indicates that the embedding is able to capture both intuitive robot-centric and object-centric task progress from visual observations. We only compare VIP and R3M here as a proxy for comparing our implicit time contrastive mechanism to the standard time contrastive learning.

The scatterplots over all tasks and camera views (Easy setting) are shown in Figure 17,18, and 19. VIP rewards exhibit much greater correlation with the ground-truth reward on its trajectories that do accomplish task, indicating that when VIP does solve a task, it is solving the task in a way that matches *human* intuition. This is made possible via large-scale value pre-training on diverse human videos, which enables VIP to extract a human notion of task-progress that transfers to robot tasks and domains. These results also suggest that VIP has the potential of *replacing* manual reward engineering, providing a data-driven solution to the grand challenge of reward engineering for manipulation tasks. However, VIP is not yet perfect in its current form. Both methods exhibit local minima where high embedding distances in fact map to lower true rewards; however, this phenomenon is much severe for R3M. On 8 out of 12 tasks, VIP at least has one camera view in which its rewards are highly correlated with the ground-truth rewards on its MPPI trajectories.

### G.5    EMBEDDING DISTANCE CURVES

In Figure 20, we present additional embedding distance curves for all methods on Ego4D and our real-robot offline RL datasets. For Ego4D, we randomly sample 4 videos of 50-frame long (see Appendix G.6 for how these short snippets are sampled), and for our robot dataset, we compute the embedding distance curves for the 4 sample demonstrations in Figure 13. As shown, on all tasks in the real-robot dataset, VIP is distinctively more smooth than any other representation. This pattern is less accentuated on Ego4D. This is because a randomly sampled 50-frame snippet from Ego4D may not coherently represent a task solved from beginning to completion, so an embedding distance curve is not inherently supposed to be smoothly declining. Nevertheless, VIP still exhibits more local smoothness in the embedding distance curves, and for the snippets that do solve a task (the first two videos), it stands out as the smoothest representation.

### G.6    EMBEDDING DISTANCE CURVE BUMPS

In this section, we compute the fraction of negative embedding rewards (equivalently, positive slopes in embedding embedding distance curves) for each video sequence and average over all video sequences in a dataset. Each sequence in our robot dataset is of 50 frames, and we use each sequence without any further truncation. For Ego4D, video sequences are of variable length. For each long sequence of more than 50 frames, we use the first 50 frames. We do not include videos shorter than 50 frames, in order to make the average fraction for each representation comparable between the

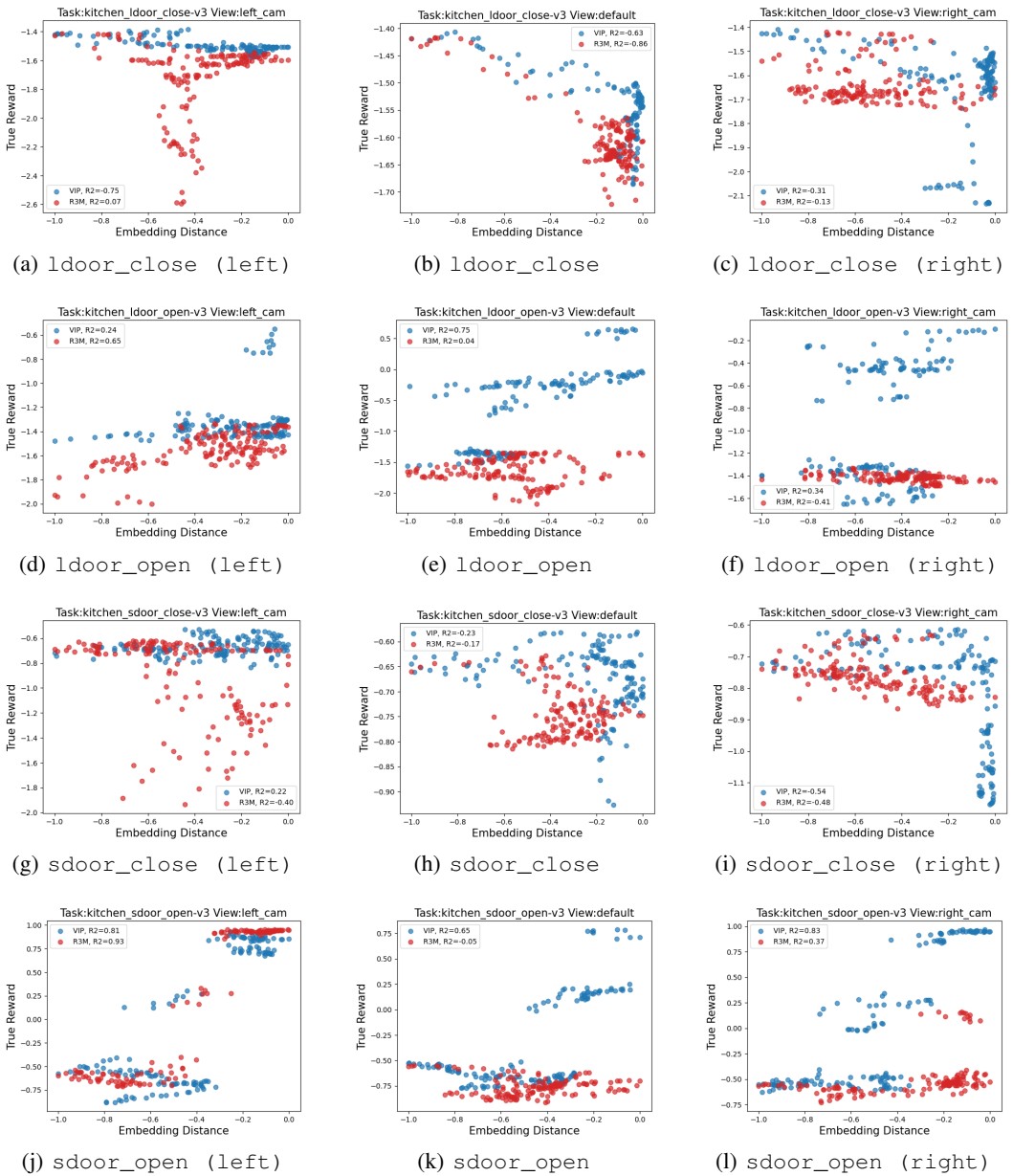

Figure 17: Embedding reward vs. ground-truth human-engineered reward correlation (VIP vs. R3M) part 1.

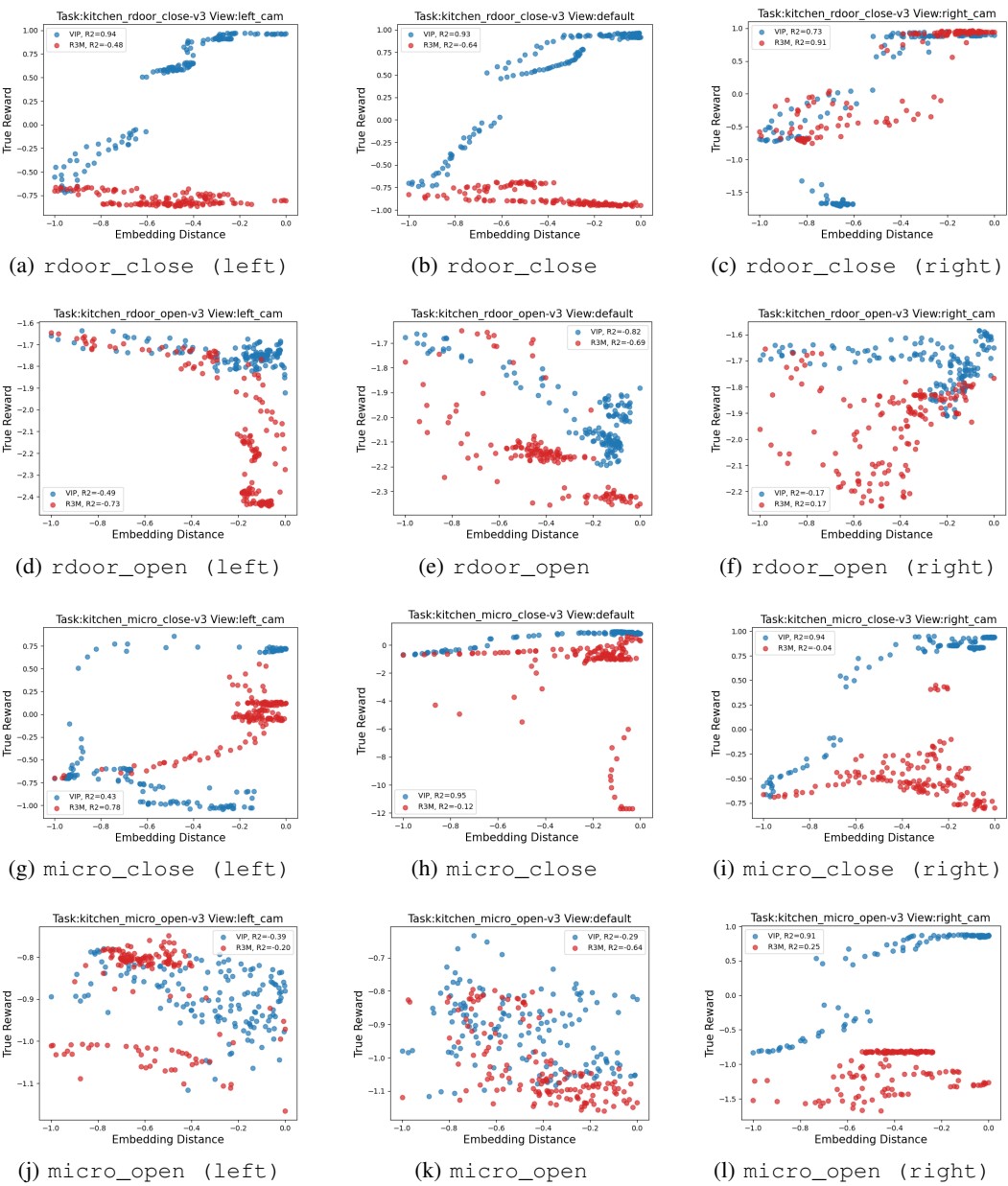

Figure 18: Embedding reward vs. ground-truth human-engineered reward correlation (VIP vs. R3M) part 2.

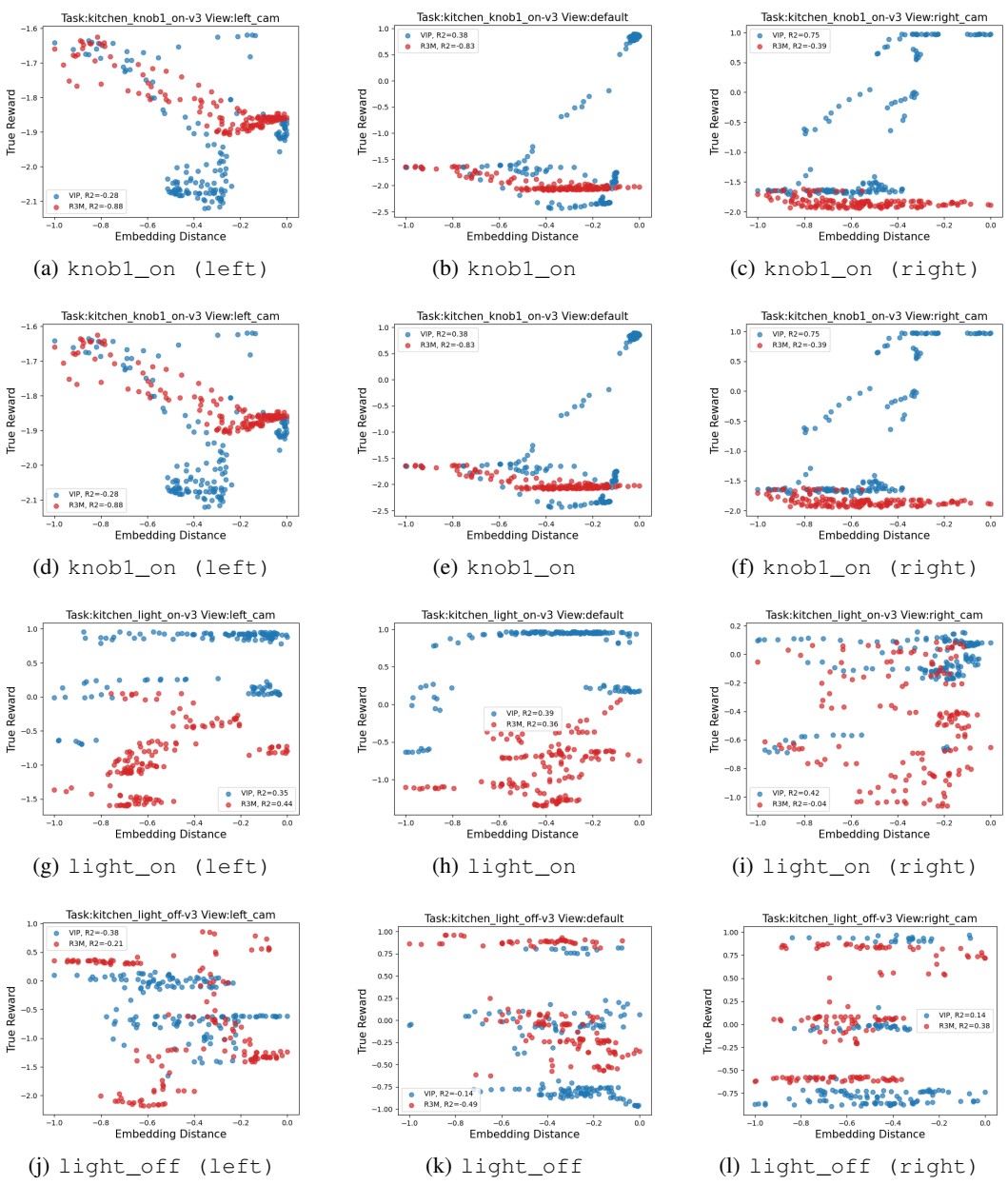

Figure 19: Embedding reward vs. ground-truth human-engineered reward correlation (VIP vs. R3M) part 3.

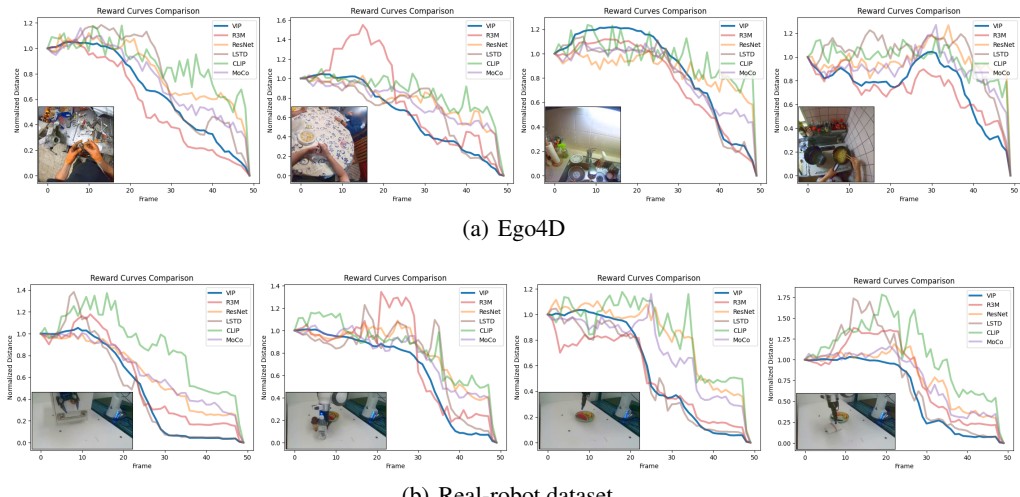

(a) Ego4D

(b) Real-robot dataset

Figure 20: Additional embedding distance curves on Ego4D and real-robot videos.

Table 5: Proportion of bumps in embedding distance curves.

| Dataset | VIP (Ours) | R3M | ResNet50 | MOCO | CLIP |
|---|---|---|---|---|---|
| Ego4D | **0.253** ± 0.117 | 0.309 ± 0.097 | 0.414 ± 0.052 | 0.398 ± 0.057 | 0.444 ± 0.047 |
| In-House Robot Dataset | **0.243** ± 0.066 | 0.323 ± 0.076 | 0.366 ± 0.046 | 0.380 ± 0.052 | 0.438 ± 0.046 |

two distinct datasets. Note that for Ego4D, due to its in-the-wild nature, it is not guaranteed that a 50-frame segment represents one task being solved from beginning to completion, so there may be naturally bumps in the embedding distance curve computed with respect to the last frame, as earlier frames may not actually be progressing towards the last frame in a goal-directed manner.The full results are shown in Table 5. VIP has fewest bumps in Ego4D videos, and this notion of smoothness transfer to the robot dataset. Furthermore, since the robot videos are in fact visually simpler and each video is guaranteed to be solving one task, the bump rate is actually *lower* despite the domain gap. While this observation generally also holds true for other representations, it notably does not hold for R3M, which is trained using standard time contrastive learning.

### G.7 EMBEDDING REWARD HISTOGRAMS (REAL-ROBOT DATASET)

We present the reward histogram comparison against all baselines in Figure 21. The trend of VIP having more small, positive rewards and fewer extreme rewards in either direction is consistent across all comparisons.

### G.8 EMBEDDING REWARD HISTOGRAMS (EGO4D)

We present the reward histogram comparison against all baselines in Figure 22. The histograms are computed using the same set of 50-frame Ego4D video snippets as in Appendix G.6. The y-axis is in log-scale due to the large total count of Ego4D frames. As discussed, Ego4D video segments are less regular than those in our real-robot dataset, and this irregularity contributes to all representations having significantly more negative rewards compared to their histograms on the real-robot dataset. Nevertheless, the relative difference ratio's pattern is consistent, showing VIP having far more rewards that lie in the first positive bin. Furthermore, VIP also has significantly fewer extreme negative rewards compared to all baselines.

## H LIMITATIONS AND FUTURE WORK

In this section, we describe limitations within the current VIP formulation and model and some potential future directions.

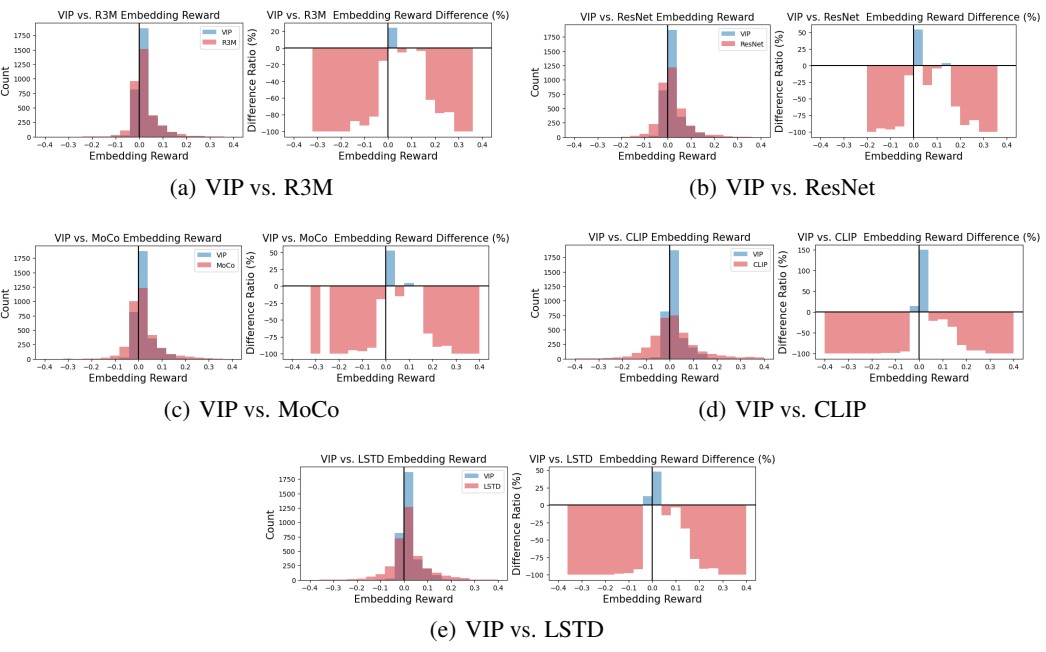

Figure 21: Embedding reward histogram comparison on real-robot dataset.

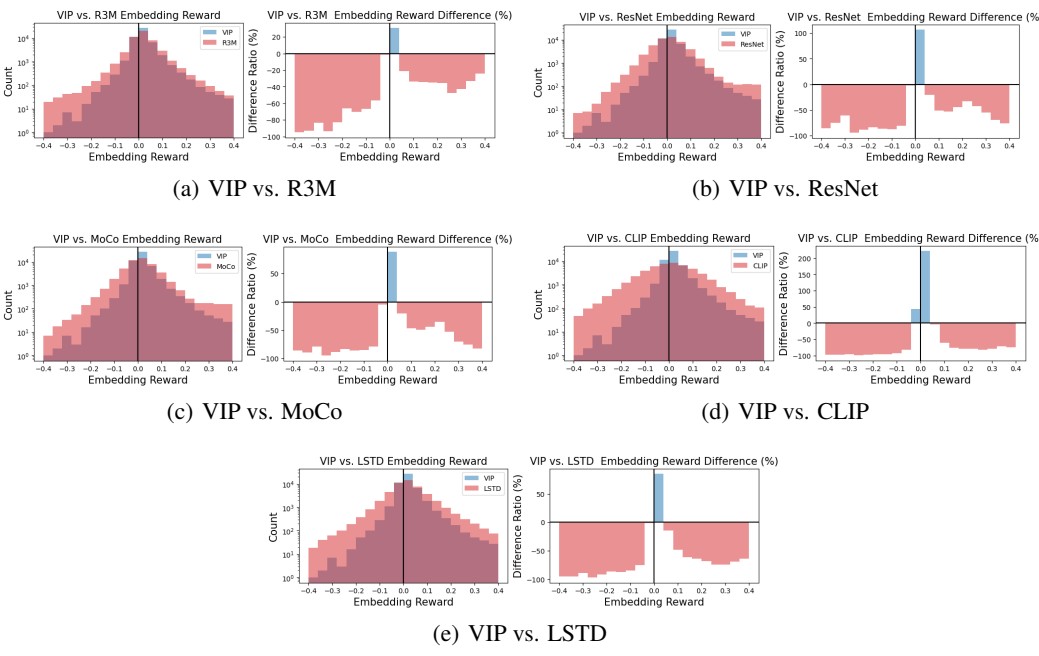

Figure 22: Embedding reward histogram comparison on Ego4D videos.

VIP is currently limited to providing rewards for tasks that can be specified via a goal image. While this encompasses a wide range of robotics tasks, many tasks cannot be fully expressed via a static image, such as ones that require following intermediate instructions and steps. Likewise, though not a strict assumption, we have only tested VIP with visual goals from the same domain (robots are not necessarily in the goal image). Extending VIP to be compatible with even more flexible and extensive forms of goals is a fruitful direction for expanding VIP's capability.

VIP current parameterizes the value function as a *symmetric* embedding distance; this assumes that the environment is reversible (i.e., it is equally easy to get from $o$ to $g$ and from $g$ to $o$), which may not hold in practice. While we did not observe this to affect practical performance, we may improve performance by parameterizing $V(o; g)$ as some distance function that supports asymmetrical structures, such as quasimetrics. Extending VIP with recent method (Wang & Isola, 2022) that can learn quasimetrics with finite data may be a fruitful future direction.

We have also used VIP only as a *frozen* visual reward and representation module to test its broad generalization capability. Better *absolute* task performance may be achieved by fine-tuning VIP on task-specific data. Exploring how to best fine-tune VIP is a promising direction for pushing VIP's limit.

Finally, we have focused on robot manipulation tasks in this work, but VIP's training objective can also be used for pre-training reward and representation for other goal-directed tasks, such as visual navigation (Savva et al., 2019). Exploring how VIP can be used to solve these other embodied AI tasks is also a promising avenue of future work.

