# OpenReview forum: "VIP: Towards Universal Visual Reward and Representation via Value-Implicit Pre-Training"
_ICLR.cc/2023/Conference — ICLR 2023 notable top 25%_

### Official Review · Reviewer_2WZ8 · 2022-10-25

**Confidence:** 3
**Correctness:** 4
**Technical Novelty And Significance:** 4
**Empirical Novelty And Significance:** 3
**Recommendation:** 8

**Clarity, Quality, Novelty And Reproducibility:**

This paper has good clarity and great quality and novelty. They also promise to release the pre-trained VIP code for reproducibility.

**Strength And Weaknesses:**

(+) The main logic of this paper is to derive a reinforcement learning target as self-supervised pre-training tasks, and use it for reinforcement learning. Therefore, it gives less domain gap in visual representation learning than in traditional CV-pretraining solutions.

(+) The method is able to compare with R3M, which also works on visual representation for robot learning. VIP does not require extra annotations such as narration and text alignment, so it is more flexible and generic.

(+) Experiments on visual trajectory optimization and online RL aggregate show VIP outperforms prior pre-trained representations, on both easy and hard cases.

(+) On a suite of real-world robot tasks, VIP can support few-shot offline reinforcement learning with smoother embedding distance curves.

(-) Although the derived pre-training target has a complicated form to implicitly optimize the value function, it is better to explain the physical meaning of each term. For example, the exponential term in log-expectation can be viewed as $$exp(||\phi(o)-\phi(g)||_2-\tilde \delta_g(o)-\gamma||\phi(o')-\phi(g)||_2)=\frac{exp((1-\gamma)||\phi(o)-\phi(g)||_2-\tilde \delta_g(o))}{exp((\gamma)(||\phi(o')-\phi(g)||_2-||\phi(o)-\phi(g)||_2))}$$
then maximizing the denominator is to maximize the one-step temporal difference, which is contradictory to the experimental observation.

**Summary Of The Paper:**

This paper proposed a self-supervised visual backbone pretraining method for reward learning in control. The key idea is to formulate representation learning from egocentric videos as an offline goal-conditioned reinforcement learning problem, whose dual-form indicated a way to implement Value-Implicit Pre-training (VIP) on unlabeled human videos. In each training iteration, VIP samples a minibatch of sub-trajectories from video, and the loss function has two components, where the first one minimizes the L2 distance of the representation of initial and goal frames, and the last one optimize the difference of L2 distance of two consecutive intermediate frames (e.g. $\cdots \gamma (F(o_{k+1})-F(o_k))$, $F$ measures the L2 distance to the goal frame embedding.). In another word, this implicit time contrastive
learning push together the feature of start/end frame, and pull away the intermediate frames implicitly.
Besides the elegant theory, the author pretrained their visual representation model on large-scale Ego4D human videos, and show the proposed pretraining idea can be combined with various on-domain solutions and significantly outperform all prior pre-trained representations on a set of simulated and real-robot tasks. A partial PyTorch code is provided in Appendix.

**Summary Of The Review:**

It is a novel idea to derive a self-supervised pre-training target from a goal-conditioned value function from RL, and the experiments are clear and supportive. I vote to accept this good paper.

---

> ### Author Response · Authors · 2022-11-09
> **Response to Reviewer 2WZ8**
>
> We thank the reviewer for their constructive comments and positive assessment of our work! Here, we respond to the questions and comments the reviewer raises. Please let us know if we can provide any additional clarifications during the discussion period to improve our score.
>
> ---
> **Question 1**: Although the derived pre-training target has a complicated form to implicitly optimize the value function, it is better to explain the physical meaning of each term. For example, the exponential term in log-expectation can be viewed as
>
> $$
> exp(||\phi(o)-\phi(g)||_2 - \tilde{\delta}_g(o) - \gamma ||\phi(o')-\phi(g)||_2 = \frac{exp((1-\gamma)||\phi(o)-\phi(g)||_2 - \tilde{\delta}_g(o))}{exp((\gamma)(||\phi(o')-\phi(g)||_2-||\phi(o)-\phi(g)||_2))}
> $$
>
> then maximizing the denominator is to maximize the one-step temporal difference, which is contradictory to the experimental observation.
>
> **Response 1**: Section 4.2 details the physical meaning of each term by re-writing Equation 3 to a log-sum-exp form (Equation 5), which can then be understood from an implicit contrastive mechanism that we develop. Please let us know if we should clarify VIP's training objective further.
>
> With regard to the given equation, we note that this decomposition is consistent with our theoretical analysis in Section 4.2. We observe that the numerator enforces the distance between o and g to not stray far away from $\tilde{\delta}_g(o)$, which is always $-1$ when $o !=g$. Because of this, while the denominator is to be maximized, it cannot grow unbounded because o’ appears in the numerator for when $(o’, o’’,g)$ is sampled to optimize the same objective. Therefore, this push-and-pull effect ensures the distance-to-goal for every intermediate observation is properly spaced, rendering a temporally coherent and smooth representation amenable for downstream reward specification.
>
> ---
>
> Please let us know if our response answers the reviewer’s questions. We are happy to provide additional clarifications to improve our score. We thank the reviewer again for their time and effort helping us improve our paper!

---

> > ### Author Response · Authors · 2022-11-17
> > **Reviewer 2WZ8 Follow-Up**
> >
> > Dear reviewer 2WZ8,
> >
> > As the discussion period is coming to a close, we wanted to check back to see whether you have any remaining questions. We would be happy to clarify further, and grateful for any other feedback you may provide.
> >
> > Thank you again for your time and feedback!
> >
> > All the best,
> > Authors

---

### Official Review · Reviewer_EXzT · 2022-10-25

**Confidence:** 3
**Correctness:** 3
**Technical Novelty And Significance:** 4
**Empirical Novelty And Significance:** 4
**Recommendation:** 6

**Clarity, Quality, Novelty And Reproducibility:**

In terms of clarity/quality:
- The paper is well written and easy to follow.
- It would have been helpful to include the rest of the related work section in the main body of the paper.

In terms of novelty:
- Overall the method seems novel, although an understanding of how VIP compares against contrastive RL would be helpful to make the exact novelty more clear.
- The bulk of the method's novelty and contribution appears to come from its ability to enable to learning observation representations from out-of-domain datasets.

In terms of reproducibility:
- The authors do not provide code, but extensive implementation details are provided in the appendix. The amount of information in the appendix should make reproduction possible, however I did not try and cannot say for certain.

**Strength And Weaknesses:**

Strengths:
- The authors conduct extensive experiments to assess the performance of VIP and to understand the benefits of each of the proposed components. In addition to conducting experiments in simulation, the authors assess the performance of VIP in a real world robot setting.
- The authors evaluate useful and relevant baselines for learning observation representations.
- The results demonstrate strong gains on prior work at the task of learning from videos of humans, which is the majority of activity-specific data available.
- Although in the appendix, the authors provide proofs alongside their proposed VIP components.
- The paper is well written.

Weaknesses:
- Large portions of the paper are punted to the appendix, include two-thirds of the related work section, two whole experiments, limitations, and future work. Despite working in a goal-conditioned RL setting, the authors punt the goal-conditioned RL related work to the appendix which makes it difficult to place the proposed method within the context of the relevant goal-conditioned RL prior work.
- It would be helpful to see how VIP performs against goal-conditioned RL where the observation encoding/representation is learned online alongside policy learning.
- For the experiments in Section 5.2, how were the hyper-parameters selected? This is important to specify to make it clear that hyper-parameters favorable to VIP were not selected.
- The need to not assess VIP under the closed-loop reinforcement learning setting is not well motivated, especially as goal-conditioned RL is deployed in such a setting.
- Online RL results for VIP (sparse) where the policy is trained using the VIP state representation and the environment's reward do not appear to be included in the paper. There is no reference to where the results can be round. Additionally, it would be nice to have more of a discussion about why it is okay for VIP's state representation to be ineffective when learning from the ground truth reward function. This is where a comparison against an online, goal-conditioned RL algorithm would be beneficial to see. Are no methods able to solve the task in the online setting? Or only those methods evaluated?
- It would be helpful to specify the length of video sequences used to train the state representations and to assess the impact of video sequence length on performance.
- It is not clear if contrastive RL (https://arxiv.org/pdf/2206.07568.pdf) is evaluated against and it would be interesting to see a discussion about the similarities and differences included in the paper.

**Summary Of The Paper:**

The paper proposed a method, VIP, to learn both an observation representation and reward using a self-supervised objective over an out-of-domain dataset. The representation learning problem is cast as offline goal-conditioned reinforcement learning for the tasks covered by the out-of-domain datasets, i.e. human activities. The reward used to learning the visual representation is 0 when the observed and goal states are the same otherwise it is -1. The reward function is selected to facilitate learning the rough distance between the current and goal frames. The representation learning objective is then cast as implicit time contrastive learning and roughly resembles InfoNCE, but without explicit negative examples. The value function for a given task is then the embedding distance between the current state and the goal state. The authors assess the impact of VIP against several representation learning baselines on trajectory optimization tasks, online RL tasks, and few-shot learning tasks to real-world robot manipulation tasks. Across experimental conditions, VIP outperforms the baselines.

**Summary Of The Review:**

Overall the paper is well written, the results demonstrate clear gains on prior work, and both empirical and theoretical evaluation is technically sound. The ability to learn representations that enables few shot learning from out-of-domain, offline data can have a large impact on the field. The largest concern with the paper is the amount of information that was punted to the related work section. The related work section and limitations are key portions of a paper to appropriately contextualize and understand the method's contribution and, therefore, that should be included in the paper's main body.

---

> ### Author Response · Authors · 2022-11-09
> **Response to Reviewer EXzT (Part 1)**
>
> We thank the reviewer for their thoughtful comments and suggestions. Here, we respond to the questions and comments the reviewer raises. First, we believe that many of the reviewers’ comments, such as the placement of goal-conditioned RL related work and baseline comparison, may come from one possible misunderstanding of VIP as studying the goal-conditioned RL problem setting. We would like to clarify upfront that this is not the case. Rather, VIP’s pre-training phase uses ideas from dual goal-conditioned RL to enable action-free value pre-training on human videos; then, the evaluation of the pre-trained of VIP presentation takes place in distinct control settings, including trajectory optimization, online RL, and offline RL, none of which is explicitly goal-conditioned (i.e., the learned policy does not condition on a goal image). The goal image is used only in constructing the task reward function (Equation 1) and is not provided to the policy. As such, VIP is loosely inspired by GCRL ideas but is not in itself a work that studies GCRL as a problem setting. We hope that this helps clarify the positioning of our paper.
>
> Now, we provide individual responses to all reviewer’s comments. Please let us know if you have lingering questions and whether we can provide any additional clarifications during the discussion period to improve the score.
>
> ---
> **Question/Comment 1**: Large portions of the paper are punted to the appendix, include two-thirds of the related work section, two whole experiments, limitations, and future work.
>
> **Response 1**: These contents are placed in the Appendix due to the strict page-limit; we will place the full related work, limitations and future work in the main text for our camera-ready, should our paper be accepted. The experiments in the Appendix (i.e., Appendix G) are not core results, but rather provide additional analysis supporting claims we made in the main paper. They are properly referenced in the main text, so we feel that it is appropriate to relegate them to the Appendix given the strict page-limit for the submission.
>
> ---
>
> **Question/Comment 2**: Despite working in a goal-conditioned RL setting, the authors punt the goal-conditioned RL related work to the appendix which makes it difficult to place the proposed method within the context of the relevant goal-conditioned RL prior work.
>
> **Response 2**: As stated above, we would like to clarify that we do not work in the goal-conditioned RL (GCRL) setting. Rather, our representation learning method VIP is derived from an offline GCRL perspective on pre-training from human videos, but all our downstream evaluation methodology is not goal-conditioned. That is, although the tasks are specified via a visual goal, the policy is not explicitly conditioned on the goal because there is only one goal that needs to be attempted; in our updated manuscript, we have further clarified this distinction at the end of Section 3. As our experiments compare different pre-trained representations fixing the policy optimization algorithm we use in each visuomotor control setting (MPPI for trajectory optimization, NPG for online RL, and RWR for offline RL), we felt the most relevant literature to our work is the pre-training representation for control literature that we do include in the main text. We have thoroughly contextualized our work in the context of prior relevant GCRL work in Appendix B; note that this section is in the Appendix only because of the strict page-limit for the main text, and we will include the full related work section in the main text for our camera-ready which allows an additional page, should our paper be accepted.
>
> ---
> **Question 3**: For the experiments in Section 5.2, how were the hyper-parameters selected? This is important to specify to make it clear that hyper-parameters favorable to VIP were not selected.
>
> **Response 3**: The hyperparameters for policy training in Section 5.2 are taken from those used in the real-robot experiment for R3M; we validated that those hyperparameters worked for BC on the simpler tasks (e.g., PushBottle and CloseDrawer) and did not make any further changes. As such, they are not selected to favor VIP, and Section 5.2 constitutes a fair comparison of the quality of the compared (R3M and VIP) visual representations and their rewards. We have added this detail in Section 5.2 in our updated manuscript.

---

> > ### Author Response · Authors · 2022-11-09
> > **Response to Reviewer EXzT (Part 2)**
> >
> > **Question 4**: The need to not assess VIP under the closed-loop reinforcement learning setting is not well motivated, especially as goal-conditioned RL is deployed in such a setting.
> >
> > **Response 4**: We do assess VIP under the closed-loop RL setting; this is our online RL experiment in Section 5.1. Our real-world offline RL setting (Section 5.2) also learns a closed-loop policy. Therefore, we do extensively evaluate VIP under closed-loop RL settings and show that it significantly outperforms all prior pre-trained representations. Please let us know if we can provide further clarifications.
> >
> > ---
> >
> > **Question 5a**: Online RL results for VIP (sparse) where the policy is trained using the VIP state representation and the environment's reward do not appear to be included in the paper. There is no reference to where the results can be round. Additionally, it would be nice to have more of a discussion about why it is okay for VIP's state representation to be ineffective when learning from the ground truth reward function.
> >
> > **Response 5a**: We note that VIP (sparse) is the cyan line at the bottom of the two online RL sub figures (second column) in Figure 4; as shown, this ablation performs quite poorly. This is to be expected because the “ground truth reward function” VIP (sparse) uses is the sparse 0-1 reward only provided when the task is successfully solved; this presents a significant exploration challenge that renders even effective state representation (e.g., VIP) ineffective, indicating the necessity of dense, shaped reward for solving these tasks.
> >
> > **Question 5b**:  This is where a comparison against an online, goal-conditioned RL algorithm would be beneficial to see. Are no methods able to solve the task in the online setting? Or only those methods evaluated?
> >
> > **Response 5b**: The default RL algorithm used and benchmarked for these tasks is the NPG algorithm we adopt in this paper; see (https://github.com/vikashplus/mjrl_dev/tree/redesign/mjrl_dev/agents/v0.1/kitchen/NPG) for the publicly released NPG training curves on these tasks using ground-truth state information and dense reward signal; in this setting, NPG is able to solve many of the tasks. Given that our goal is to compare various pre-trained representations and rewards, we fix the underlying RL algorithm to be the NPG algorithm, and study solving these tasks from raw visual inputs and without crafted dense rewards.
> >
> > **Question 5c**: It would be helpful to see how VIP performs against goal-conditioned RL where the observation encoding/representation is learned online alongside policy learning.
> >
> > **Response 5c**: Since our downstream representation evaluation is not goal-conditioned and the standard algorithm for this benchmark is NPG, one appropriate and straightforward comparison that does both representation and policy learning simultaneously is an end-to-end version of NPG trained from scratch, where the visual representation is learned online alongside policy learning. This will take more than a full day for training for each task due to graphics rendering and forward/backward operations through a ResNet architecture, and given that we have 36 tasks, this comparison would be difficult to complete in the rebuttal period. Instead, we compare against NPG trained with **ground-truth state** and sparse-reward, which serves as an upper bound for end-to-end NPG because the perfect state representation is already provided. We train this variant of NPG on all 12 tasks with 3 seeds each, and find that it cannot solve any task in both Easy and Hard settings. This suggests that learning the representation alongside the policy will most likely do poorly as well, highlighting the necessity of pre-training both visual representation and reward for making progress on these tasks and VIP’s unique effectiveness in doing so. We have included a sentence in Section 5.1 summarizing this finding.
> >
> > ---
> > **Question 6**: It would be helpful to specify the length of video sequences used to train the state representations and to assess the impact of video sequence length on performance.
> >
> > **Response 6**: We train the VIP representation using the default pre-truncated version of Ego4D, consisting of clips ranging from 10-150 frames; we have added this information to our updated manuscript (Appendix D.1). It may indeed be interesting to investigate training VIP using various fixed clip lengths, but this requires re-training and re-evaluating the model each time. Given the size of the Ego4D dataset and the time required, this may be hard to do during the rebuttal period. We will try to include these results for camera-ready and future revisions.

---

> > > ### Author Response · Authors · 2022-11-09
> > > **Response to Reviewer EXzT (Part 3):**
> > >
> > > **Question 7**: It is not clear if contrastive RL (https://arxiv.org/pdf/2206.07568.pdf) is evaluated against and it would be interesting to see a discussion about the similarities and differences included in the paper.
> > >
> > > **Response 7**: We have discussed this work (Eysenbach et al., 2022) in our related work in Appendix B; as stated, we will move the entire related work section to the main text for our camera-ready. At a more technical level, this method cannot be applied to our pre-training setting because it requires action labels for training the contrastive Q-function, and human videos do not contain action labels.  Likewise, the similarity is only superficial in that both Eysenbach et al and our work interprets some GCRL algorithm as some form of contrastive learning. Their work depends on a particular form of reward function in order for goal-conditioned Q-learning to be interpreted as InfoNCE contrastive learning, whereas VIP’s implicit time contrastive learning is derived from a dual form of an offline GCRL objective, (2) admits a novel time contrastive mechanism altogether, and (3) holds for any choice of reward function. Therefore, this prior work and VIP differ significantly in  motivation, technical derivation, and applications.
> > >
> > > ---
> > > **Question 8**: The authors do not provide code, but extensive implementation details are provided in the appendix. The amount of information in the appendix should make reproduction possible, however I did not try and cannot say for certain.
> > >
> > > **Response 8**: We would like to clarify that the code is included in our supplementary material; a PyTorch-like pseudocode is also provided in Appendix D.3. Please let us know if you are unable to access the code supplementary material.
> > >
> > > ---
> > >
> > > **Citations**:
> > > Eysenbach, et al. Contrastive learning as goal-conditioned reinforcement learning.
> > >
> > > Please let us know if our response adequately answers the reviewer’s questions regarding the positioning, novelty, experiments, and the reproducibility of the paper. We are happy to provide additional clarifications to improve our score. We thank the reviewer again for their time and effort helping us improve our paper!

---

> > > > ### Author Response · Authors · 2022-11-17
> > > > **Reviewer EXzT Follow-Up**
> > > >
> > > > Dear reviewer EXzT,
> > > >
> > > > As the discussion period is coming to a close, we wanted to check back to see whether you have any remaining questions. We would be happy to clarify further, and grateful for any other feedback you may provide.
> > > >
> > > > Thank you again for your time and feedback!
> > > >
> > > > All the best,
> > > > Authors

---

### Official Review · Reviewer_DfUc · 2022-10-25

**Confidence:** 4
**Correctness:** 4
**Technical Novelty And Significance:** 3
**Empirical Novelty And Significance:** 4
**Recommendation:** 8

**Clarity, Quality, Novelty And Reproducibility:**

The paper is well written, striking a good balance between technical detail and understanding for a broader audience.  The work is novel to the best of my knowledge, despite some similarities between VIP and other recent papers showing that goal reaching RL is related to contrastive objectives (and those approaches are cited and discussed in the paper).  The code is provided and although I have not combed it in fine detail, at first glance it seems clean and approachable.

**Strength And Weaknesses:**

The paper presents very compelling empirical results, and a clear derivation and description of the method.  The method is novel, and takes large steps toward solving very difficult problems in reinforcement learning and robotics.  The literature is carefully and extensively surveyed, and the contributions of VIP are clearly laid out in contrast to the relevant prior methods.  The method is also straightforward, and can be summarized by a three step algorithm, which suggests it can be easily reproduced.

**Summary Of The Paper:**

The paper presents Value-implicit pretraining (VIP) which is a method to train representations from images for use in robotic reinforcement learning.  The method uses contrastive learning to generate smooth representations of trajectories, where the start and goal states of trajectories are encouraged by the loss to be close in the representation space and intermediate observations are encouraged to interpolate smoothly between the two.  Because the representation is trained to interpolate between start end goal states, the authors argue that this interpolation is equivalent to a value function when the reward is a goal-reaching reward.  The representation can thus be used to assess distance to the goal state, and thus can be used as a dense reward.  The paper shows strong empirical results generalizing representations learned in egocentric human video data sets to robotic tasks.

**Summary Of The Review:**

This is a very strong paper with an interesting approach to a difficult and important problem.  The empirical results are strong and I think this is a clear accept.

---

> ### Author Response · Authors · 2022-11-09
> **Response to Reviewer DfUc**
>
> We thank the reviewer for their time and positive assessment of our work! Please let us know if we can provide any additional clarifications during the discussion period.

---

### Official Review · Reviewer_s2zb · 2022-10-25

**Confidence:** 4
**Correctness:** 3
**Technical Novelty And Significance:** 3
**Empirical Novelty And Significance:** 3
**Recommendation:** 6

**Clarity, Quality, Novelty And Reproducibility:**

- Clarity: The proposed method and evaluation are clearly clarified.
- Quality: The paper is well-written and structured.
- Novelty And Reproducibility: The paper solves the problem of reward designing in complex robotic control domains by leveraging out-of-domain human demonstrations and goal frame instructions, contrastively learning a visual representation that implicitly encodes value function. The idea is interesting, significant, and new. Details of method implementation are clear and thus make it reproducible.

**Strength And Weaknesses:**

Strength:
1. The motivation of learning visual reward functions from human demonstrations without action annotations and then generalizing it to unseen robotic control tasks is intriguing.
2. The proposed dual-objective goal-conditioned contrastive learning enables more accurate and smooth embedding distance through full demonstration.
3. The paper is well-written and easy to follow.
Weakness:
1. In Section 5.2, Table 1 shows that VIP in-domain representation learning is worse than Scratch-BC and even has no task progress due to overfitting. Is VIP easily affected by the quality of the demonstration? What if sub-optimal or not diverse enough, will it hinder RL update?
2. Does VIP exceed in-domain imitation learning metrics? It would be more convincible if you compare your metric with IL baselines trained on in-domain demonstrations in Kitchen benchmarks, for example, behavior transformer, Implicit Behavioral Cloning.
3. Is your model able to handle non-Markovian demonstrations where different actions concatenate sequentially, instead of single clips?
4. On your project website, robot motion in MPPI trajectory seems to be fluctuating and unstable, is there any explanation or improvement for this problem?

**Summary Of The Paper:**

This paper is in the realm of generalizable reward learning and the aim is to learn value-discriminative visual representation from a diverse set of human manipulation demonstrations by contrasting conditioned on goals. The learned representation is used for constructing reward as similarity in embedded space between adjacent steps. The proposed time contrastive method differs from standard unidirectional contrastive learning, it attracts initial and final frames while repelling intermediate frames, thus presenting smoothly decreased embedding distances. The method is evaluated on trajectory optimization, online RL, and real-world offline RL tasks to demonstrate effectiveness of the visual representation over other pretraining baselines.

**Summary Of The Review:**

The paper proposes an efficient method for visual reward and representation learning from out-of-domain data, which highly benefits downstream tasks. I would say it is an improvement to existing work but can be a good inspiration on how to contrastively learn a representation of demonstrations.

---

> ### Author Response · Authors · 2022-11-09
> **Response to Reviewer s2zb**
>
> We thank the reviewer for their thoughtful comments and suggestions! Here, we respond to the questions and comments the reviewer raises. Please let us know if you have lingering questions and whether we can provide any additional clarifications during the discussion period to improve the score.
>
> **Question 1**: In Section 5.2, Table 1 shows that VIP in-domain representation learning is worse than Scratch-BC and even has no task progress due to overfitting. Is VIP easily affected by the quality of the demonstration? What if sub-optimal or not diverse enough, will it hinder RL update?
>
> **Response 1**: The reason VIP (in-domain) performed poorly is not due to the quality of the offline dataset but rather the size of the offline dataset (~20 trajectories = 1000 transitions). This comparison demonstrates that training a value function is extremely challenging in the few-shot offline RL regime we study, and highlights the importance and necessity of pre-training a visual representation and reward on large corpus of out-of-domain data (i.e., human videos) as in VIP (pre-trained). Finally, we note that Scratch-BC still performs poorly, and the best approach is our pre-trained VIP-RWR. Because VIP-RWR leverages good reward signals provided by the VIP representation, it is actually more robust to the quality of the demonstrations. This is corroborated by its higher performance on all tasks, where mixed-quality demonstrations are collected.
>
> ---
>
> **Question 2**: Does VIP exceed in-domain imitation learning metrics? It would be more convincible if you compare your metric with IL baselines trained on in-domain demonstrations in Kitchen benchmarks, for example, behavior transformer, Implicit Behavioral Cloning.
>
> **Response 2**: Our simulated Kitchen benchmark (Section 5.1) evaluates trajectory optimization and online RL, for which the reward signal needs to be generated by the representation itself and no demonstration is provided for learning. Therefore, imitation learning algorithms do not apply in these settings. However, in our real-robot experiments (Section 5.2, Table 1), we do compare against a fully in-domain IL baseline, BC (Scratch), and show that VIP-based offline RL substantially outperforms it, signaling that end-to-end IL is not effective in this few-shot offline RL regime (similar finding has been reported in Nair et al., 2022). Finally, we note that algorithms such as Behavior Transformer or Implicit Behavior Cloning are alternative IL algorithms to the simple MSE-based BC approach we adopt in this work; they are compatible with VIP and are not direct comparisons to it. As our goal is to compare different pre-trained visual representations, we have fixed the imitation learning algorithm to the simple choice of MSE, which is a standard choice in the literature (Nair et al., 2022, Parisi et al., 2022).
>
> ---
>
> **Question 3**: Is your model able to handle non-Markovian demonstrations where different actions concatenate sequentially, instead of single clips?
>
> **Response 3**: Our real-robot demonstrations for real-world few-shot offline RL are collected by human kinesthetic teaching; it is well known that human demonstrations are not necessarily Markovian (Mandlekar et al., 2021). Yet, our VIP (RWR) excelled in this challenging and novel problem setting.
>
> ---
>
> **Question 4**: On your project website, robot motion in MPPI trajectory seems to be fluctuating and unstable, is there any explanation or improvement for this problem?
>
> **Response 4**: Since we re-compute the best action according to MPPI at each rollout step,  temporal smoothness of consecutive actions is not explicitly encouraged, resulting in fluctuating actions. As such, this behavior is an artifact of the optimization approach (occurs with VIP and all other baselines) and simulation dynamics and not a weakness of our pre-trained representation. In contrast, we do observe that VIP with other policy optimization approaches produce smooth trajectories. For example, in our real-world experiments (Section 5.2), VIP reward-weighted regression (RWR) produces smooth and successful trajectories on all four evaluation tasks (sample videos are on the project website).
>
> **Citations**:
>
> Nair, et al. R3M: A universal visual representation for robot manipulation. CORL, 2022.
>
> Parisi, et al. The unsurprising effectiveness of pre-trained vision models for control. ICML, 2022.
>
> Mandlekar, et al, What matters in learning from offline human demonstrations for robot manipulation. CORL, 2021.
>
> ---
>
> Please let us know if our response answers the reviewer’s questions. We are happy to provide additional clarifications to improve our score. We thank the reviewer again for their time and effort helping us improve our paper!

---

> > ### Author Response · Authors · 2022-11-17
> > **Reviewer s2zb Follow-Up**
> >
> > Dear reviewer s2zb,
> >
> > As the discussion period is coming to a close, we wanted to check back to see whether you have any remaining questions. We would be happy to clarify further, and grateful for any other feedback you may provide.
> >
> > Thank you again for your time and feedback!
> >
> > All the best,
> > Authors

---

### Author Response · Authors · 2022-11-09
**Rebuttal Revision Has Been Posted**

Dear Reviewers, AC, and PC,

Our revised paper has been posted. For clarity, we highlighted all changes in red. The main changes include:

1. A clarification on the policy learning setting at the end of Section 3.
2. A discussion on a new state-based, sparse-reward baseline in Section 5.1.

We thank all reviewers for their time and effort helping us improve our paper, and we look forward to discussing our paper with you further during the discussion period.

Best,

Authors

---

### Decision · Program_Chairs · 2023-01-20

**Decision:**

Accept: notable-top-25%

**Justification For Why Not Higher Score:**

Although a very strong paper, I feel the scope of the work should still be considered incremental (a significant increment). Not ground-breaking.

**Justification For Why Not Lower Score:**

As mentioned, it is a strong paper with both theoretical support and thorough empirical studies.

**Metareview: Summary, Strengths And Weaknesses:**

About generalizable reward learning and its aim is to learn value-discriminative visual representation from a diverse set of human manipulation demonstrations by contrasting conditioned on goals. The method uses contrastive learning to generate smooth representations of trajectories, where the start and goal states of trajectories are encouraged by the loss to be close in the representation space and intermediate observations are encouraged to interpolate smoothly between the two. The paper shows strong empirical results generalizing representations learned in egocentric human video data sets to robotic tasks.

+ compelling empirical results on challenging benchmarks;
+ The method is also straightforward, and can be summarized by a three-step algorithm, which suggests it can be easily reproduced.

A few technical concerns raised by reviewers and the authors addressed them to a certain extent.

**Note From Pc:**

if the above contains the word "oral" or "spotlight" please see: "oral" presentation means -> notable-top-5% and "spotlight" means -> notable-top-25%. As stated in our emails, we are disassociating presentation type from AC recommendations

**Summary Of Ac-Reviewer Meeting:**

N/A